# Attentional shifts bias microsaccade direction but do not cause new microsaccades
Baiwei Liu ✉, Zampeta-Sofia Alexopoulou & Freek van Ede ✉

Brain circuitry that controls where we look also contributes to attentional selection of visual contents outside current fixation, or content within the spatial layout of working memory. A behavioural manifestation of this contribution comes from modulations in microsaccade direction that accompany spatial attention shifts. Here, we address whether such modulations come about because attention shifts trigger new microsaccades or whether, instead, spatial attention only biases the direction of ongoing microsaccades that would have been made whether or not attention was also shifted. We utilised an internal-selective-attention task that has recently been shown to yield robust spatial microsaccade modulations and compared microsaccade rates following colour retrocues that were carefully matched for sensory input, but differed in whether they invited an attention shift or not. If attention shifts trigger new microsaccades then we would expect more microsaccades following attention-directing cues than following neutral cues. In contrast, we found no evidence for an increase in overall microsaccade rate, despite robust modulations in microsaccade direction. This implies that shifting spatial attention biases the direction of ongoing microsaccades without changing the probability of microsaccade occurrence. These findings help to explain why microsaccades and visual-spatial shifts of attention are often correlated but not obligatorily linked.

Goal-driven visual-spatial attention can be directed not only overtly – by looking directly at task-relevant visual information – but also covertly, by attending to relevant information outside of current fixation[1,2] or to information held internally within working memory[3–5]. Ample prior research has demonstrated that brain circuitry that is involved in overt eye-movement control also contributes to the deployment of covert visual-spatial attention (e.g., refs. [6–10]).

One particular example of this contribution comes from the study of microsaccades, a class of fixational eye-movements that occur even during attempted fixation (e.g., refs. [11–15]). In particular, it has been demonstrated how the direction of fixational microsaccades can be modulated by the deployment of spatial attention to peripherally attended locations, in the absence of large eye-movements to these locations (refs. [16–26]; but see also refs. [27–29] to which we return in our discussion). Building on this earlier work, we recently uncovered similar microsaccade biases when directing attention to memorised visual contents held within the spatial layout of working memory (e.g., refs. [30–36]). This provides a useful model system for studying the link between microsaccades and spatial attention because in this set-up there

is no incentive for large eye-movements as the objects of attention are in mind.

When considering the link between microsaccades and visual-spatial attention, a relevant question is whether shifting spatial attention leads to spatial microsaccade modulations because attention shifts themselves trigger microsaccades that follow the direction of the attention shift. Alternatively, shifting spatial attention may not trigger any microsaccades directly, but merely bias the direction of *ongoing* microsaccades – i.e., naturally occurring microsaccades that would have been made whether or not spatial attention was also shifted.

These alternative scenarios have often remained tacit in the literature. Yet, disambiguating these scenarios is likely to be informative for our understanding of the links between spatial attention, microsaccades, and upstream oculomotor brain circuitry. Foremost, the answer to our question may help delineate the probabilistic[26,27,32,37] versus deterministic[19] nature of the link between microsaccades and covert shifts of visual-spatial attention. If spatial attention only biases the direction of ongoing microsaccades, without triggering new microsaccades, then this would explain a probabilistic link whereby spatial attention can also be shifted without a

Institute for Brain and Behavior Amsterdam, Department of Experimental and Applied Psychology, Vrije Universiteit Amsterdam, Amsterdam, The Netherlands.
✉e-mail: b.liu@vu.nl; freek.van.ede@vu.nl

concomitant microsaccade (corroborating the findings reported in refs. 32,37). In addition, our findings may guide future neurophysiological studies targeting the role of upstream oculomotor brain circuitry, such as the superior colliculus – a brain structure implicated in both selective covert spatial attention[8,38,39] and microsaccade generation[40,41]. For example, based on our results, we may derive distinct hypotheses that spatial attention adds new activity to the pool of superior-colliculus neurons (triggering new microsaccades) or instead mainly acts by biasing the balance of neuronal activity (yielding only a bias in the *direction* but not the *rate* of microsaccades).

To address whether spatial attention shifts add new microsaccades or bias ongoing ones, we compared microsaccade rates following attentional cues that were carefully matched for sensory input, but that differed whether they invited a spatial attention shift or not. We note how the comparison between informative (attention-directing) and neutral cues was also available in refs. 17,20 though in these studies the spatial biasing by voluntary shifts of spatial attention itself was weak and the cues were not perfectly matched, making it hard to address the question that we put central here. Indeed, it has been shown how bottom-up cue features can also modulate microsaccades[42,43], thus contaminating interpretation if not carefully matched between informative and neutral cues. Here we carefully matched informative and neutral cues and studied microsaccade-rate modulations in the context of an internal selective-attention task that we recently established as a powerful model system because it yields robust spatial modulations in microsaccades by spatial shifts of attention (here, directed within the spatial layout of visual working memory)[30–32,35,36,44]. Our logic was straightforward: if voluntary shifts of visual-spatial attention trigger new microsaccades – that account for the observed spatial modulation in microsaccade direction – then overall microsaccade rates should be higher following attention-directing than following neutral cues. In contrast, if spatial attention merely biases the direction of ongoing microsaccades, then we should see a biasing effect on microsaccade direction *without* a concomitant increase in overall microsaccade rate.

## Methods

Experimental procedures were reviewed and approved by the local Ethics Committee at the Vrije Universiteit Amsterdam. Each participant provided written informed consent before participation and was reimbursed 10 euros/hour. The study was not preregistered.

### Participants

Twenty-five healthy human volunteers from Vrije Universiteit Amsterdam participated in the study (age range: 18–44; 5 men and 20 women; 25 right-handed; 5 corrected-to-normal vision: 1 glasses and 4 lenses). Gender was self-reported. Sample size of 25 was determined a-priori based on previous publications from the lab with similar experimental designs that relied on

the same outcome measure (e.g., refs. 30,31,34). One participant was excluded for all analyses due to chance-level performance.

### Stimuli and procedure

To investigate microsaccade modulations by voluntary visual-spatial shifts of attention, we employed an internal selective-attention task (Fig. 1) for which we previously established robust spatial modulations in microsaccades (see e.g., refs. 30,32,35,36,44). In short, participants encoded two visual items into working memory in order to compare the orientation of either memory item to an upcoming test stimulus. In a random half of the trials, a retrocue presented during the retention interval informed which memory item would become tested by briefly changing the colour of the central fixation marker to match the colour of the target memory item (attention-directing cue). In the other half of the trials, we also presented a colour cue, but this time cue colour did not match either item in memory, and hence did not invite a shift of attention to either memory content (neutral cue).

Each trial began with a brief (250 ms) encoding display in which two bars (size: 2° × 0.4° visual angle) appeared at 5° to the left and right of the fixation. After an initial retention delay of 1250 ms, the fixation dot (0.07° radius) changed colour for 1000 ms serving as a retrocue that prompted participants to select the colour-matching target item in memory. After another retention delay of 500 ms, the test display appeared in which a target bar appeared at the centre of the screen. The target bar matched the colour of the target memory item but was rotated between 10 to 20 degrees clockwise or counter-clockwise from its original orientation. Participants were required to report the tilt offset of the test stimulus using the keyboard ('j' for clockwise, 'f' for counter-clockwise). Participants received feedback immediately after the response by a number ("0" for wrong, or "1" for correct) appearing for 250 ms slightly above the fixation dot. After the feedback, inter-trial intervals were randomly drawn between 500 and 1000 ms.

In the experiment, bars could be four potential colours: green (RGB: 133, 194, 18; measured luminance: 77 candela/$m^2$), purple (RGB: 197, 21, 234; measured luminance: 24 candela/$m^2$), orange (RGB: 234, 74, 21; measured luminance: 31 candela/$m^2$), and blue (RGB: 21, 165, 234; measured luminance: 53 candela/$m^2$). The fixation marker was grey (RGG: 128 128 128; measured luminance: 36 candela/$m^2$) by default and changed into either of the four abovementioned colours to serve as a cue. For each participant, bars were always chosen from a random subset of three of these colours. In each encoding display, bars were randomly assigned two distinct colours from the available colour pool and two distinct orientations ranging from 0° to 180° with a minimum difference of 20° between each other.

To address our central question whether visual-spatial shifts of attention trigger additional microsaccades, we included both attention-directing and neutral retrocues. In half the trials, the retrocue colour matched either

**Fig. 1 | Internal selective attention task with attention-directing and neutral colour retrocues.** Participants encoded two visual items into working memory in order to later compare the orientation of one of the items to a test stimulus that was tilted 10 or 20 degrees clockwise or counter-clockwise relative to the colour-matching memory item. During the delay, the colour of the central fixation dot changed colour serving as a cue. In a random half of the trials, the retrocue matched the colour of either item in working memory, informing with 100% reliability that this item would become tested. In the other trials, the cue also involved a colour change, but this time the colour did not match either item in working memory. Colours of memory items and cues were counterbalanced, such that a physically identical colour cue would be attention directing in some trials while neutral in other trials.

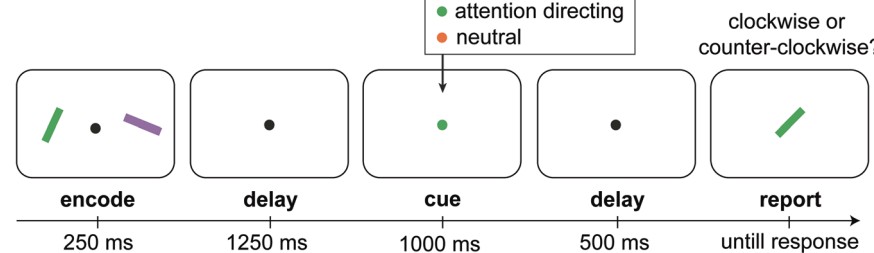

memory item, inviting a shift of attention to the to-be-tested memory content. Participants were encouraged to use these informative, attention-directing retrocues that were 100% valid. In the other half of the trials, the retrocue was drawn from either colour that was *not* in the encoding display, thus not inviting a shift of attention among the contents of working memory. In these trials, participants would know which memory item was the target memory item only upon the presentation of the coloured test stimulus.

In total, the study consisted of 2 sessions, each containing 5 blocks of 48 trials, resulting in a total of 480 trials. Both conditions (attention-directing cues and neutral cues) were randomly intermixed within each block as were attention-directing cues to left and right memory items, resulting in 240 attention-directing trials (120 directing spatial attention to the left memory item, 120 directing spatial attention to the right memory item), and 240 neutral trials. At the start of the experiment, participants practiced the task for 48 trials. We did not include practice trials in our analyses. The study was not preregistered.

### Eye-tracking acquisition and pre-processing
Using an EyeLink 1000 with a sampling rate of 1000 Hz, we continuously tracked gaze along the horizontal and vertical axes from the right eye. The eye tracker was placed ~5 cm in front of the monitor and ~65 cm away from the eyes. Before recording, we calibrated the eye tracker through the built-in calibration and validation protocols from the EyeLink software. Gaze data was originally recorded in .edf format and was converted to .asc format to be further analysed after recording.

We analysed the data in Matlab with help of the Fieldtrip analysis toolbox[45] and custom code. To clean the data from blinks, we marked blinks by detecting clusters of zeros in the time-series eye data. To eliminate residual blink artifacts, all data from 100 ms before to 100 ms after the detected blink clusters were interpolated. After blink removal, data were epoched relative to retrocue onset.

### Saccade detection
To detect saccades, we employed a velocity-based method that we established previously[32], and that builds on other established velocity-based methods for microsaccade detection (e.g., ref. [17]). Since the items in the current experiment were always horizontally arranged (i.e., left and right), our current analyses focused exclusively on the horizontal channel of the eye data. Note that although we only use horizontal data to detect saccades, we previously confirmed the validity and sensitivity of this approach for our task-setup by comparing this method to a well-established method (as described in[17]) that considered both horizontal and vertical gaze (see ref. [32] for the relevant comparison).

We first calculated the gaze velocity by taking the distance between temporally successive gaze positions. Then, to reduce noise, we smoothed velocity in the temporal dimension with a Gaussian-weighted moving average filter with a 7-ms sliding window (using the built-in function "smoothdata" in MATLAB). We then identified the first sample when the velocity exceeded a trial-based threshold of 5 times the median velocity as the onset of a saccade. To avoid counting the same saccade multiple times, we imposed a minimum delay of 100 ms between successive saccades. Saccade magnitude and direction were calculated by estimating the difference between pre-saccade gaze position (−50 to 0 ms before threshold crossing) vs. the post-saccade gaze position (50 to 100 ms after threshold crossing). Finally, depending on saccade direction (left/right) and the side of the cued memory item (left/right), we labelled every detected saccades as "toward" or "away". To enable a direct comparison of the spatial modulation in saccades following attention-directing vs. neutral cues, we also separated "toward" and "away" saccades following neutral cues that did not invite an attention shift to either memory item. For this, we artificially classified saccades as "toward" or "away" based on the memory item that would *eventually* be tested – but that could not yet be known in the period of interest after the cue.

After identifying and labelling the saccades, we quantified the time courses of saccade rates (in Hz) using a sliding time window of 100 ms, advanced in steps of 1 ms. To map the size of the modulated saccades (without setting an arbitrary saccade-size threshold), we additionally decomposed saccade rates into a time-size representation (as in refs. [32],[36],[44]), showing the time courses of saccade rates, as a function of the saccade size. We did this twice: once using a linear and once using a logarithmic spacing of saccade size. For linear saccade-size sorting, we used successive magnitude bins of 0.5 visual degrees in steps of 0.05 visual degree. For logarithmic saccade-size sorting, we used 100 steps between 0.1 and 6.3 degrees, and used bin widths equal to the size at which each bin was centred.

To look at the influence of ongoing saccade rate preceding the cue, we also sorted trials into low-rate and high-rate trials preceding the cue. For this, we extracted the total number of saccades in the second preceding the cue and defined low-rate trials as trials in which the number of saccades was equal or smaller than the median, and high-rate trials as trials in which the number of saccades was larger than the median.

To directly quantify the number of saccades during the attentional window of interest we averaged saccade rates in the 200–600 ms window after cue onset. This window was set a-priori based on our prior study that revealed this to be the critical window after cue onset in which we found more microsaccades toward vs. away from the memorised location of the cued memory target (see ref. [32]).

### Statistical analysis
For the analysis of the behavioural data, we compared informative-cue to neutral-cue conditions using a Linear Mixed Model, with default settings as implemented in Jamovi (The jamovi project (2021). Jamovi. (Version 1.6) [Computer Software]). For RT, we first log-transformed the data. For binary accuracy scores (incorrect, correct), we used a logistic regression.

To evaluate the reliability of statistical patterns we observed in the time-series gaze data, we employed a cluster-based permutation approach[46]. This method is ideal for evaluating significance while circumventing the problem of multiple comparisons.

We first acquired a permutation distribution of the largest cluster size by randomly permuting the trial-average data at the participant level 10,000 times and identifying the size of the largest clusters after each permutation. To obtain the probability (*P* value) of the clusters observed in the original data, we calculated the proportion of permutations for which the size of the largest cluster after permutation was larger than the size of the observed cluster in the original, non-permuted data. The permutation analysis was conducted using Fieldtrip with default clustering settings. That is, after a mass univariate *t*-test at a two-sided alpha level of 0.05, we identified and grouped the adjacent same-signed data points that were significant and then defined cluster size as the sum of all *t*-values in the cluster.

In addition to the cluster-based permutation approach that considered the full time range, we also extracted the data over the pre-defined 200–600 ms window after cue onset (based on ref. [32]). We compared the relevant conditions using paired-sample *t*-tests and report Cohen's d as a measure of effect size, in all cases where the reported data did not violate the assumption of normality. In one case, normality could not be assumed, and we performed a Wilcoxon rank sum test instead. Data distributions are visualised by including the individual-participant data in all bar graphs. In addition, we employed Bayesian statistics on the data over this extracted time window, as implemented in Jamovi (The jamovi project (2021). Jamovi. (Version 1.6) [Computer Software]), with default priors ($r = 0.707$). This enabled us to quantify evidence in favour of the null (BF01) that informative retrocues do not increase total microsaccade rates in comparison to the control condition with neutral retrocues.

### Reporting summary
Further information on research design is available in the Nature Portfolio Reporting Summary linked to this article.

## Results

Human volunteers performed a selective-attention task in which selective voluntary spatial attention was directed to one of two visual representations held within the spatial layout of working memory (Fig. 1). In half of the trials, a central colour cue directed spatial attention to the colour-matching visual item in working memory (attention-directing cues). In the other half of the trials, we also presented a colour cue but this time the cue did not match either memory item and therefore did not invite a shift of spatial attention (neutral cues). This served as the critical control condition to assess whether shifting spatial attention triggered new microsaccades: in both cases a central colour cue appeared but only in the former condition a goal-directed shift of spatial attention could be made.

As a roadmap to our results, we first report behavioural performance to confirm that participants used the cue when it invited a shift of attention to the target memory item. We then outline the spatial modulation of micro-saccade direction when cues directed spatial attention to either the left or right memory item. Having established the above, we finally turn to our key question whether this spatial microsaccade modulation is driven by the addition of new, spatial-attention-driven microsaccades or, instead, by a biasing of ongoing microsaccades that would have been made anyway. For this, we compared overall microsaccade rate between trials with attention-directing cues versus neutral cues. The logic was as follows: if spatial-attention shifts add new microsaccades then overall rate should increase following attention-directing compared to neutral cues. In contrast, if spatial-attention shifts merely bias the direction of ongoing microsaccades then we should observe similar rates following attention-directing and neutral cues.

### Informative attentional cues during working memory improve ensuing memory-guided behaviour

Before turning to our main eye-movement results, we first turn to the behavioural performance in our task, to confirm that informative (attention-directing) cues were used by participants (c.f.[4]). As shown in Fig. 2, participants were both more accurate (Fig. 2a) and faster (Fig. 2b) in trials with attention-directing versus neutral cues. This was corroborated statistically using linear mixed modelling, both for accuracy ($\beta = 0.613$, SE = 0.064, z = 9.64, $p < 0.001$) and for reaction time ($\beta = 0.197$, SE = 0.021, t(23) = 9.31, $p < 0.001$).

### Attentional cues during working memory lead to a robust spatial modulation in microsaccades

Having confirmed that participants used the informative (attention-directing) cues to improve performance, we next assessed how informative

cues – that directed attention to memory items that had been presented to the left or right at encoding – modulated the direction of microsaccades, consistent with the spatial deployment of attention within the spatial layout of visual working memory.

Building on our prior studies[30–32,35,36,44] as well as related studies deploying external covert-attention tasks[16,17,19,21,22,24,26], we observed robust biasing of saccade directions as a function of whether cues directed spatial attention to memory items that had been presented to the left or to the right of fixation at encoding (Fig. 3a). When statistically evaluating the full time course, we observed more saccades toward than away from the memorised location of the cued item (black horizontal line in Fig. 3a; cluster $P = 0.0001$). Likewise, when zooming in on the a-priori defined time window from 200–600 ms after the cue (based on ref. 32), we observed a highly robust modulation (Fig. 3a, bar graph; t(23) = 4.568, $p = 0.0001$, d = 0.933; 95% CI = [0.101, 0.267]) with more saccades toward than away from the memorised location of the cued item. Later in our results section we show that this modulation is driven by saccades in the microsaccade range, as revealed by a complementary analysis and visualisation that we present in Fig. 5.

For the purpose of enabling a direct comparison between attention-directing and neutral cues, we also considered saccade rates following neutral cues (Fig. 3b) whose colour did not match either memory item and therefore did not invite a shift of spatial attention to either memory item. For this, we artificially classified saccades as "toward" or "away" based on the memory item that would *eventually* be tested – but that could not yet be known in the period of interest after the cue. Because neutral cues did not inform which item would become tested, it was re-assuring that after neutral cues we observed no difference in the number of "toward" and "away" saccades (Fig. 3b, bar graph; t(23)= −0.285, $p = 0.778$, d = −0.058, 95% CI = [−0.052, 0.040]). Figure 3c shows the overlay of the spatial biasing of microsaccades (toward minus away) following cues that did vs. did not invite a spatial shift of attention to either memory item. A direct comparison showed significantly larger spatial biasing following attention-directing vs. neutral cues, both when comparing the full time courses (cluster $P = 0.005$ and 0.027) or the spatial bias averaged over the a-priori defined window of interest (Fig. 3c, bar graph; t(23) = 3.733, $p = 0.001$, d = 0.762, 95% CI = [0.085, 0.296]; Wilcoxon rank sum test: W = 268, $p < 0.001$, r = 0.787).

When reflecting on these findings, please note how, in our task, the spatial biasing of saccades following attention-directing cues must reflect a shift of spatial attention to the colour-matching memory item that was held within the spatial layout of working memory. It cannot reflect sensory processing of the cue nor anticipation of the probe, as both cue and probe

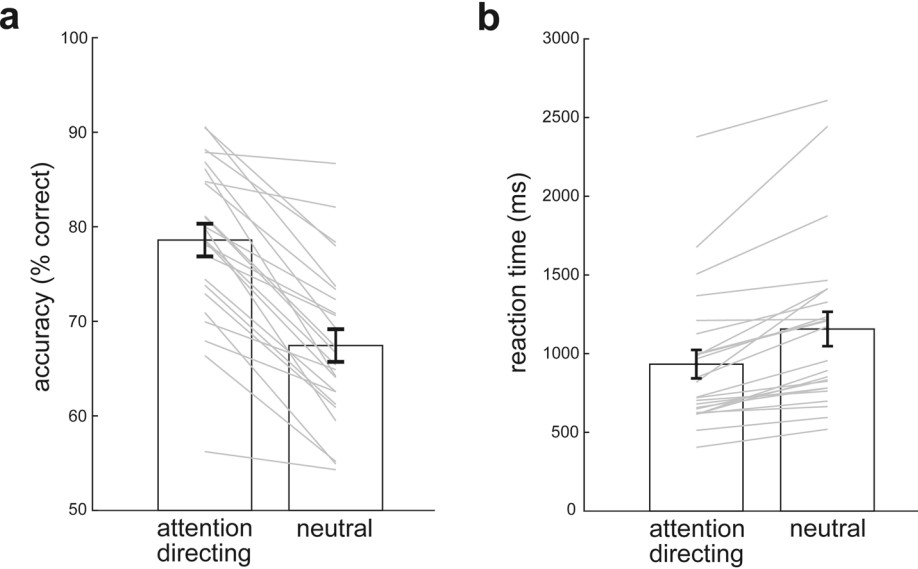

**Fig. 2 | Behavioural performance confirms participants used the cue when possible. a** Task accuracy in trials with attention-directing and neutral cues. The y-axis in panel a starts at 50%, reflecting chance-level accuracy. **b** Reaction times in trials with attention-directing and neutral cues. Error bars indicate ± 1 SEM calculated across participants (*n* = 24). Grey lines denote individual participants. For reaction times, we calculated median reaction times across trials in light of the skewed nature of reaction-time distributions.

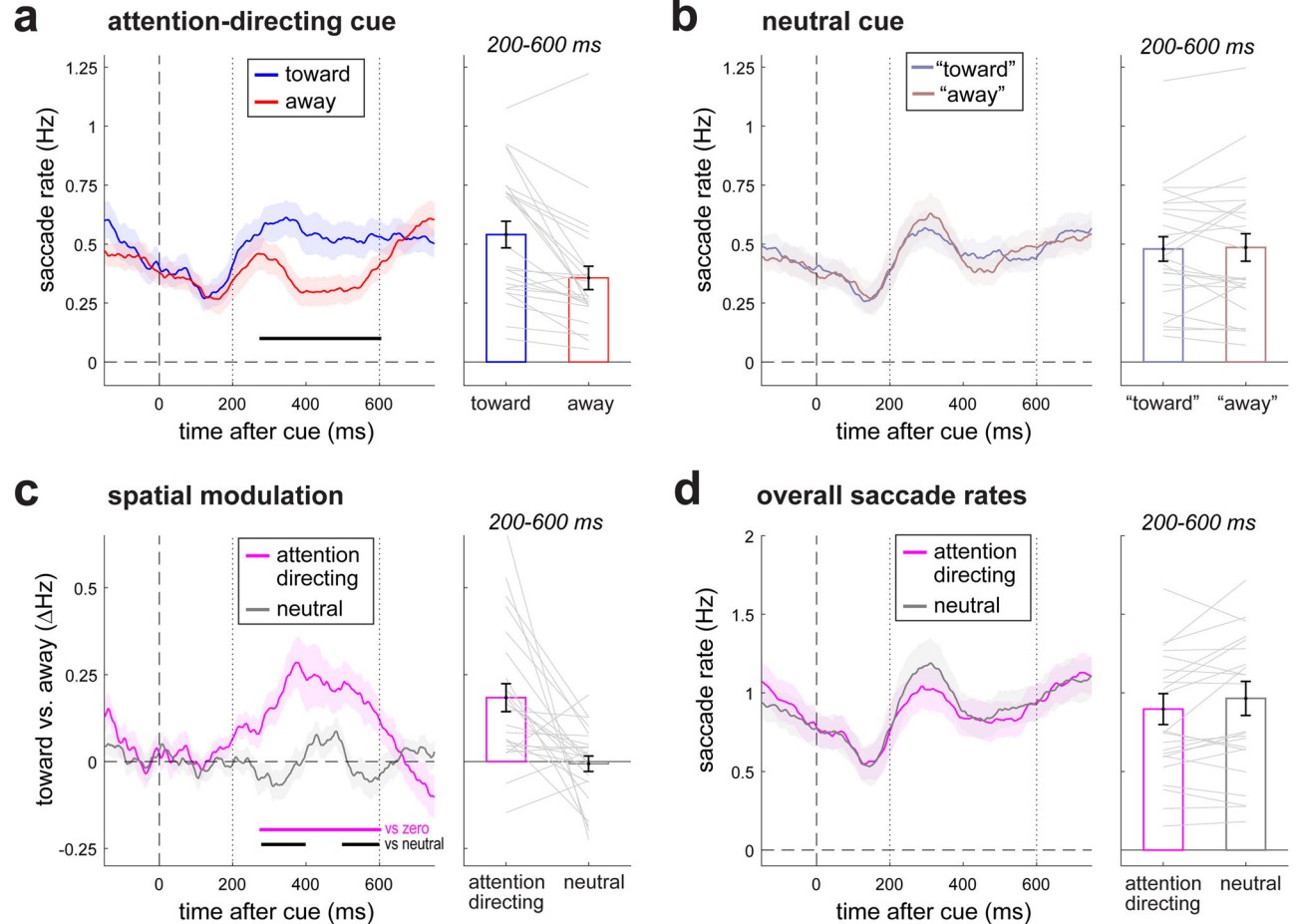

**Fig. 3 | Internal selective attention modulates the direction of microsaccades without changing overall microsaccade rate. a** Saccade rates after cue onset in trials with attention-directing cues for saccades in the direction of the memorised location of the cued memory item (toward) and in the opposite direction (away). **b** Saccade rates after cue onset in trials with neutral cues. Note how neutral cues do not inform which item will become tested later on, rendering no meaningful difference between "toward" and "away" in this post-cue interval of interest (toward and away we here defined based on which item would later be tested, and serve only to showcase as a reference and to confirm the absence of any spatial bias following neutral cues).

**c** Spatial saccade-direction bias after cue onset following attention-directing and neutral cues. **d** Overall saccade rates after cue onset following attention-directing and neutral cues. Note how saccade rates in panel d are higher than in **a, b**, given that overall saccade rates include both toward and away saccades. Horizontal lines denote significant difference clusters following cluster-based permutation[46]. Bar graphs show data averaged over the a-priori defined window from 200–600 ms after cue onset (based on ref. [32]). Error bars indicate ± 1 SEM calculated across participants (*n* = 24). Grey lines denote individual participants.

were always presented centrally (i.e., left and right were exclusively defined in the memorised visual space).

### The modulation of microsaccades during shifts of spatial attention is not driven by the addition of new microsaccades

Having established a robust spatial biasing of microsaccades following attention-directing cues (Fig. 3a, c), we now turn to the central question of the current study: whether the above-described spatial modulation of microsaccades is driven by the addition of new, spatial-attention-driven, microsaccades or whether this spatial modulation is driven by a directional biasing of ongoing microsaccades that would have been made anyway. For this, we compared overall microsaccade rates following informative, attention-directing, cues versus following neutral cues. We reasoned that if spatial shifts of attention introduce new microsaccades then the overall rate should increase following attention-directing compared to neutral cues.

Overall saccade rates following attention-directing and neutral cues are shown in Fig. 3d. In contrast to the above prediction, and in contrast to the clear differences in spatial modulation between conditions (Fig. 3c), we found no evidence for an increase in overall microsaccade rate following attention-directing cues compared to following neutral cues (that were matched in terms of bottom-up sensory stimulation). In fact, if anything, we

found a slight decrease in overall microsaccade rate following attention-directing cues (Fig. 3d), though this did not reach significance – neither when considering the full-time axis (no significant clusters), nor when zooming in on the a-priori defined window of interest (t(23) = −1.971, *p* = 0.061, d = −0.402, 95% CI = [−0.1395, 0.0034]). Moreover, a Bayesian analysis showed strong evidence in favour of the null, i.e., for the *absence* of an increase in total microsaccade rates following informative vs. neutral cues (BF01 = 12.3). This dissociation whereby spatial shifts of attention modulate saccade direction (Fig. 3c) but not rate (Fig. 3d) implies that spatial-attention shifts do not generate new microsaccades.

### The lack of an overall increase in microsaccade rate is not due to a ceiling effect

We report a clear spatial modulation in microsaccade direction (Fig. 3c) without a concomitant increase in overall microsaccade rate (Fig. 3d). This makes clear that spatial modulations in microsaccades *can* exist without the addition of new (spatial-attention-triggered) microsaccades. At the same time, it remains possible that our findings are specific to cases where microsaccade rates are already close to ceiling, such that no new microsaccades can be added. To address this possibility, we sorted our trials by the rate of saccades in the second preceding the cue (using a median split),

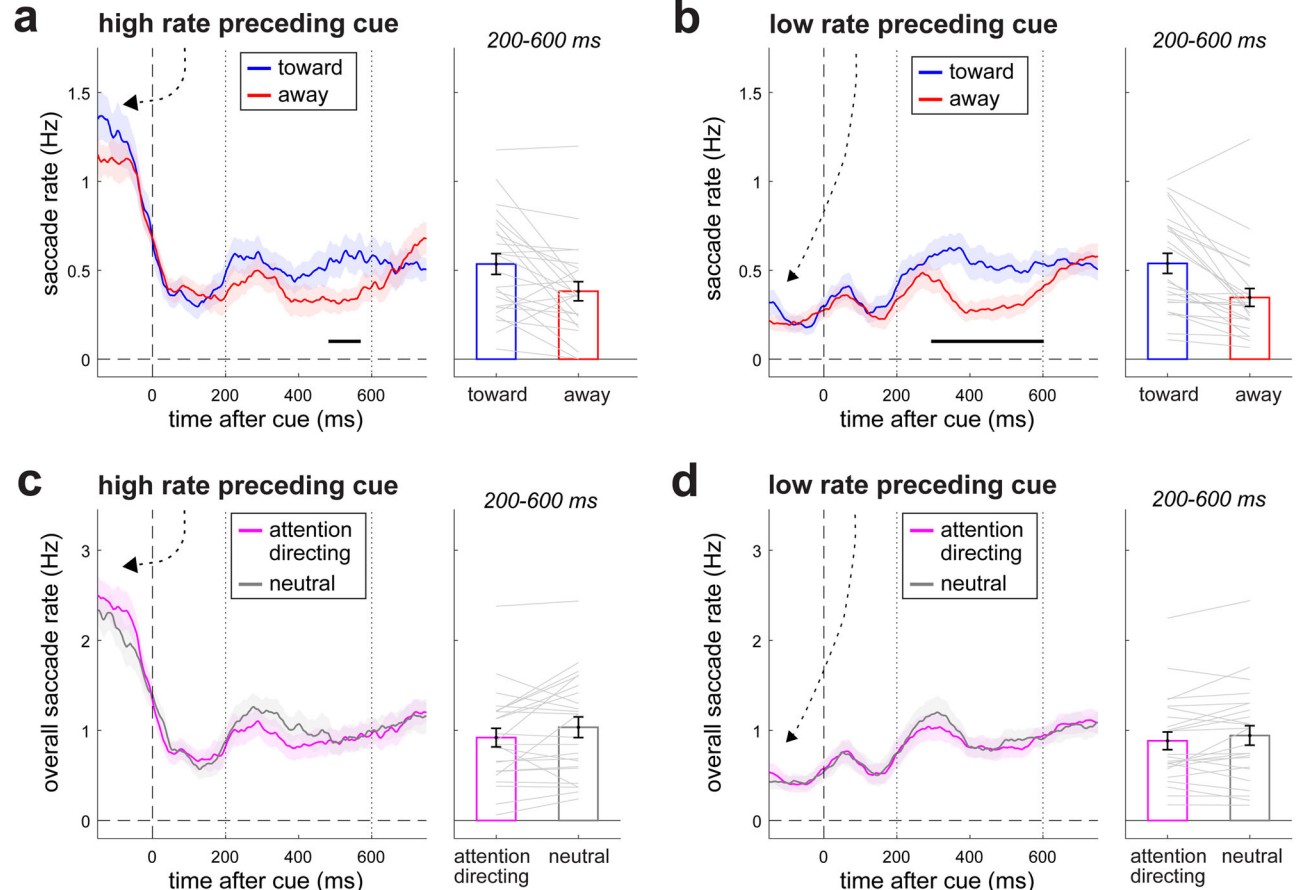

**Fig. 4 | The lack of an overall rate increase is not due to a ceiling effect. a** Toward and away saccades following attention-directing cues in trials with high saccade rates preceding cue onset (median split). **b** Toward and away saccades following attention-directing cues in trials with low saccade rates preceding cue onset. **c** Overall saccade rates after cue onset following attention-directing and neutral cues in trials with high saccade rates preceding cue onset. **d** Overall saccade rates after cue onset following attention-directing and neutral cues in trials with low saccade rates preceding cue onset. Error bars indicate ± 1 SEM calculated across participants (n = 24). Conventions as in Fig. 3.

yielding "high-rate trials" (were ongoing saccade rate was closer to a putative ceiling) and "low-rate trials" (where ongoing saccade rate was further from ceiling). If the lack of an overall increase in the number of microsaccade following attention-directing vs. neutral cues is due to a ceiling effect, we should find this lack of an overall increase in rate only in the high-rate, but not the low-rate trials. In contrast, we observed the same qualitative pattern of results after the cue regardless of preceding saccade rate (Fig. 4). While pre-cue saccade rates confirmed our trial sorting, we found that high-rate (Fig. 4a, c) and low-rate (Fig. 4b, d) trials each showed a clear spatial modulation following attention-directing cues (Fig. 4a, b; cluster $P = 0.032$, 0.0001; bar-graph comparisons: $t(23) = 4.564$, $p = 0.0001$, d = 0.932, 95% CI = [0.105, 0.279]; $t(23) = 3.302$, $p = 0.031$, d = 0.674, 95% CI = [0.057, 0.249], for high-rate and low-rate trials respectively) without an increase in overall saccade rate (Fig. 4c, d no clusters found; bar-graph comparisons: $t(23) = -1.699$, $p = 0.103$, d = $-0.347$, 95% CI = [$-0.132$, 0.013]; $t(23) = -1.902$, $p = 0.078$, d = $-0.388$, 95% CI = [$-0.239$, 0.010]). This shows that our key finding – a lack of an overall increase in rate, despite a clear spatial modulation – is not restricted to cases where saccade rate is already close to ceiling.

### Attentional biasing of saccades is driven by saccades in the microsaccade range

Having delineated our central finding of a dissociable effect of spatial attention on saccade direction vs. saccade rate, we finally demonstrate that the reported saccade modulations are driven by saccades in the microsaccade range, as we had already alluded to above. To demonstrate this, we

repeated our key analyses as a function of saccade size and visualised our results using both a linear and logarithmic spacing of saccade size. As can be seen in Fig. 5 the vast majority of saccades occurred in the microsaccade range, below 1 degree visual angle. This is perhaps not surprising given that in the post-cue time period of interest, there was nothing on the screen apart from the fixation dot.

When directly comparing toward and away saccades following attention-directing cues (Fig. 5a, right panel), we found that the spatial modulation too was confined to the microsaccade range, replicating our previous findings[30,32,36,44]. Note how the items were centred at 5 degrees visual angle at encoding, as indicated by the dashed horizontal line in the top-right panel in Fig. 5. This makes clear how the reported spatial modulation is *not* driven by participants looking back *at* the memory items' original locations (as in "looking-at-nothing"; e.g., refs. 47–51), but rather by microsaccades *toward* this location.

When considering overall microsaccade rate (Fig. 5b), we again found a clear prevalence of saccades below 1 degree, following both attention-directing and neutral cues. Consistent with our main findings in Fig. 3, we again found no evidence for more saccades in this microsaccade range following attention-directing cues (Fig. 5b, right), despite the clear spatial modulation that we observed following these cues.

### Discussion

We observed robust modulations in microsaccade direction, with more microsaccades toward versus away from the memorised location of a cued visual memorandum (replicating our previous work, e.g.,

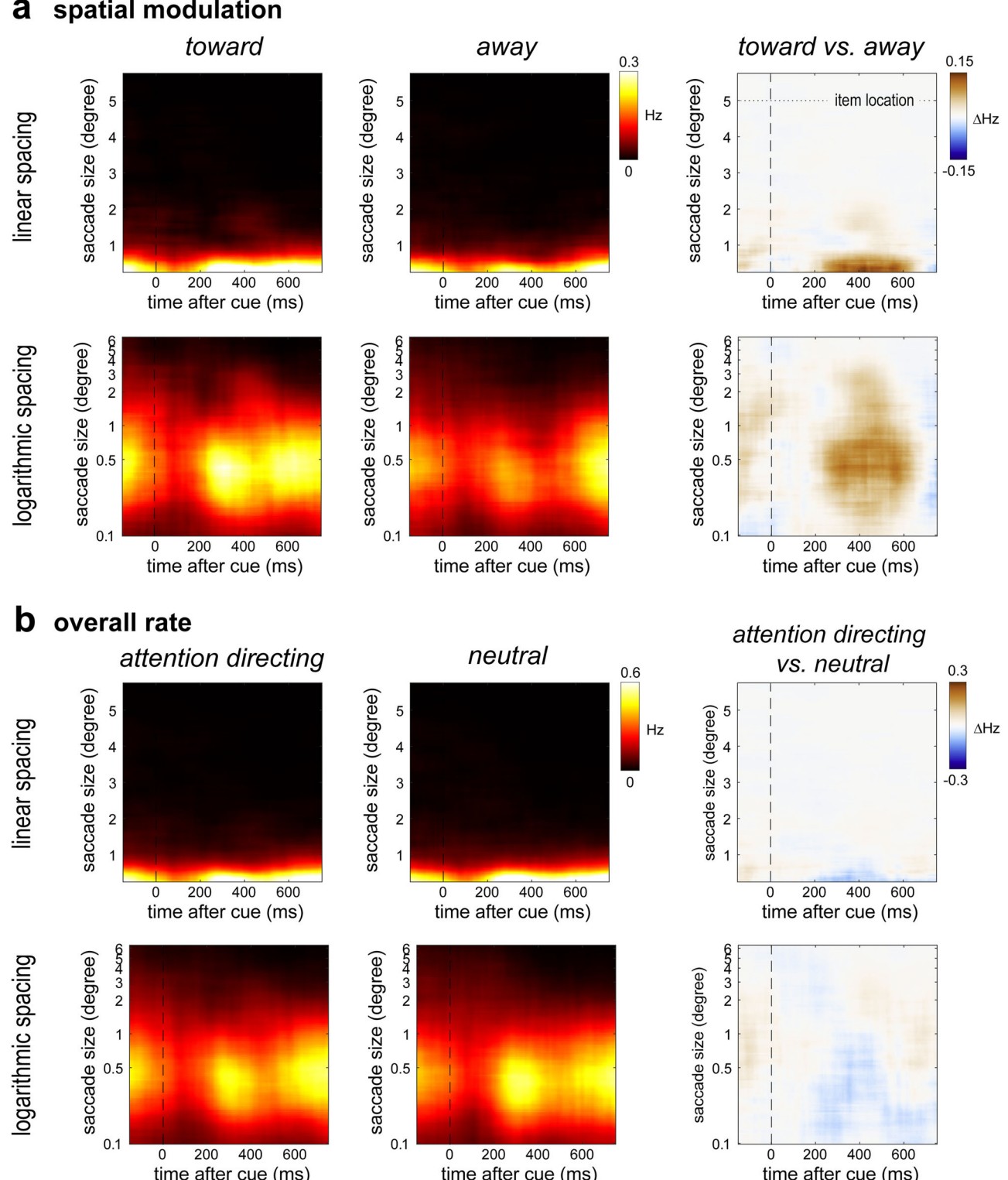

**Fig. 5 | Attentional biasing of saccades is driven by saccades in the microsaccade range. a** Saccade rates as a function of saccade size (y axes) and time after attention-directing cues (x axes) for toward saccades (left), away saccades (middle), and their difference (toward minus away; right). **b** Overall saccade rates as a function of saccade size and time after cue onset for trials with attention-directing cues (left), neutral cues (middle), and their difference (attention-direction minus neutral; right). Data were binned and visualised as a function of linearly spaced saccade sizes (top) and logarithmically spaced saccade sizes (bottom). During encoding, items were centred at ± 5 degrees to the left and right of fixation.

refs. 30,32,35,36,44). Our aim was to assess whether this spatial modulation is driven by the addition of new microsaccades that are triggered directly by a spatial shift in attention. To this end, we compared overall microsaccades rates following attention-directing cues to a control condition with neutral cues that did not invite any shift of spatial attention, but that were matched in sensory properties otherwise. Our data showed no evidence for an increase in overall microsaccade rate following attention-directing cues (if anything we observed a slight, albeit non-significant decrease). This lack of a

**Fig. 6 | Hypothetical schematic of how attention may bias microsaccade direction without changing total microsaccade rate.** Data show hypothetical activity in neural populations coding for the left and right memory items as well as for left and right microsaccades, somewhere in the oculomotor system. Population activity is assumed to determine the probability of a microsaccade, hence microsaccade rate. When attention is deployed to the right (middle column), activity in the right population is boosted, yielding a net increase in total activity across the population, yielding more microsaccades. When divisive normalisation[55,56] is introduced (right column), the attention-triggered increase in activity in the right population is normalised across the total population. This re-balancing of activity across the left and right populations yields a bias in microsaccade direction without a net increase in microsaccade rate.

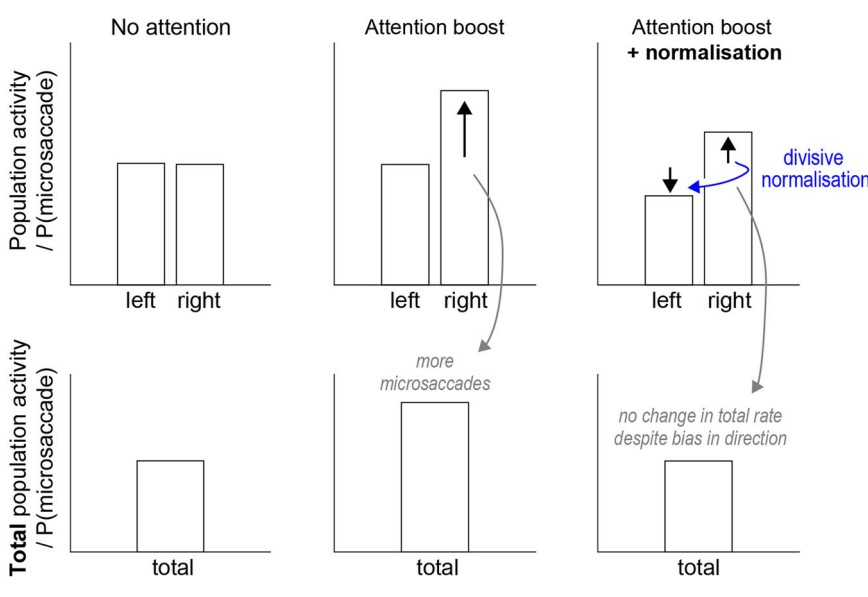

rate increase in the face of a clear directional modulation implies that the modulation of microsaccades during visual-spatial shifts of attention is not driven by the injection of "new" microsaccades. Instead, these data suggest that spatial attention merely *biases* the direction of ongoing microsaccades that would have been made whether or not attention was also shifted. In other words, shifting attention does not change the probability that a microsaccade will occur, but it does change the probability where a microsaccade will go – *if* one will be made. This pattern of results helps to understand why the link between microsaccades and attention shifts is merely probabilistic and not obligatory: because spatial attention shifts only bias ongoing microsaccades, the link between spatial attention shifts and microsaccades is "at the mercy" of there being a microsaccade in the first place.

Previous work has suggested that microsaccades are correlated with, but not necessary for spatial attentional shifts (e.g., refs. 26,27,32,37). Our findings are consistent with, and help to appreciate or explain, this probabilistic nature of this link between microsaccades and spatial attention. Because attention shifts themselves do not trigger microsaccades, it is possible to have attention shifts without a peripheral trace in the form of a microsaccade. Instead, only when attention shifts are made in the presence of an (already planned) microsaccade, will we observe a correlation between microsaccade direction and the direction of the covert or internal shift of attention. It is noteworthy, however, that we here studied microsaccades when participants are explicitly cued to voluntarily shift attention. Complementary work has shown how spontaneous microsaccades – made in the absence of volitional shifts of attention – may themselves trigger performance and neural modulations that are typically associated with attention shifts[52–54]. Whether and how spatial attention can become decoupled from such spontaneous microsaccades, or whether attention may inevitably follow in the case of spontaneous microsaccades, remains an interesting question not addressed by the current study.

By studying microsaccades as an accessible peripheral signature of upstream oculomotor brain circuitry, our findings have implications for our understanding of the links between spatial attention and the oculomotor system. For example, it has previously been established that the superior colliculus, besides regulating saccades and microsaccades, also plays a key role in shifting covert visual-spatial attention (e.g., refs. 8,38,39). The spatial biases we observed in the microsaccade direction are consistent with this. At the same time, the lack of an overall increase in microsaccade rate suggests that spatial attention shifts do not necessarily lead to a net increase in activity in the oculomotor system. Instead, we speculate based on our data that spatial attention may re-balance the distribution of activity across the

oculomotor system, as schematically illustrated in Fig. 6. When spatial attention is deployed, neuronal populations representing the attended memory item may increase their activity, but this increase in activity is normalised across the total population via divisive normalisation[55,56]. Such re-balancing would predict that the distribution of activity may vary depending on spatial attention, while overall population activity – and with this the probability of generating a microsaccade and thus microsaccade rate – remains on par. Such a scenario, though purely hypothetical at present, may provide a simple explanation for our finding of not more microsaccades, but instead the same number of microsaccades that go more in the attended direction. We wish to make clear however that our intention here is not to prove this specific model. Rather, we merely aim to provide one possible way to make sense of our findings in which attention clearly modulates the relative rates of toward vs. away microsaccades but without a net increase in total microsaccade rate. While computational modelling of our findings is beyond the scope of the current work, it will be important in future work to integrate these ideas with existing models of microsaccade generation (such as (Engbert et al.[57]; Engbert[23])). In this light, it is also important to note how our findings are consistent with the autonomous saccade timing assumption in the popular SWIFT model of saccade generation[57], which posits that cognitive processes may inhibit the timer, but not generate saccades directly.

A recent study[28] found little to no evidence for a directional biasing of microsaccades to a spatially attended visual stimulus, despite clear behavioural and neural benefits of attention. A critical difference with our study is that the authors did not consider *shifts* of spatial attention following a cue, but rather *sustaining* spatial attention to either of two visual targets that remained fixed throughout a block of trials. It is conceivable that microsaccade biases may be particularly sensitive to shifts of covert visual-spatial attention, without necessarily also tracking the process of sustaining covert visual-spatial attention after this initial shift[58]. Another set of complementary studies focused on microsaccades following exogenous capture of attention to a peripheral cue. These studies have typically reported microsaccades biases that look away from the cued location (e.g., refs. 20,23), rather than the observation of more toward microsaccades that we reported here, and that previous studies also reported following voluntary spatial attention cues. Whether such microsaccades biases that occur in the opposite direction also reflect biasing of ongoing microsaccades or rather the addition of new microsaccades remains an interesting question for future work.

At least two early studies on microsaccade biases by visual-spatial attention also included neutral cues[17,20]. While their data thus allowed the same key comparison as we targeted here, there are several relevant

differences. One relevant difference is that in these studies the employed endogenous cues showed only weak directional biasing of microsaccades[20]. In comparison, here, we observed a clear bias; building on our prior work using the same overall task set-up (e.g., refs. 30–32,34–36,44). It was only in the presence of this clear spatial modulation that our question was of interest: whether we could "explain" this spatial modulation by the addition of new microsaccades. Second, in aforementioned work, the authors did not fully match informative (attention-directing) and neutral cues in terms of bottom up cue features. This complicates a direct comparison given that differences in bottom-up cue features can also lead to differences in microsaccades[42,43]. In contrast, we always used the same colour cues, and counterbalanced whether specific colours were attention-directing or neutral. Third, we here included a relevant additional analyses to show that the lack of an overall rate increase is not due to a ceiling effect. Finally, we unique compared microsaccades following attention-directing versus neutral cues in the context of an internal selective attention task in which spatial attention was directed internally to the contents of working memory – thus extending prior observations from the domain of perception to the domain of working memory.

As alluded to above, a primary reason we resorted to a task in which attention was directed within the spatial layout of visual working memory was pragmatic: because we[30,32,35,44] as well as others[59,60] have recently shown how this task yields particularly robust spatial modulations in microsaccades. In future studies it will of course be interesting to assess similar questions in complementary settings such as with spatial attention directed externally to anticipated or already present visual stimuli; with spatial attention directed to stimuli in other sensory modalities; with attention directed to multiple items/locations in concert, and so on. Moreover, in addition to our focus on spatial modulations in microsaccades during visual-spatial shifts of attention, it will be interesting to expand our question to non-spatial forms of attention that have also been linked to microsaccades (e.g., refs. 61–63). It is further noteworthy how non-spatial forms of attention, such as temporal expectation and/or alertness, may sometimes also attenuate microsaccade rates (e.g., refs. 64,65). Indeed, a transient increase in alertness may possibly also account for the transient drop in microsaccades ("oculomotor freezing") that we observed after cue onset.

## Limitations
Our central conclusion hinges on the combination of two findings: the *presence* of a robust spatial modulation in combination with the *absence* of a modulation in total microsaccade rates (in comparison to our neutral condition). While the latter involves a "null result", it is important to consider this null result in relation to the clear spatial modulation. Our spatial modulation demonstrates clear sensitivity of our task and analysis pipeline to microsaccade-rate modulations, as our dependent variable of interest. Yet, in the same dataset, we find no evidence for a concomitant net increase in microsaccade rate. It is this combination of results that makes our specific null finding compelling – showing that the spatial bias is unlikely accounted for by a scenario in which attention shifts themselves trigger accompanying microsaccades. This was further corroborated by a Bayesian analysis showing evidence in favour of no net increase in microsaccade rate following attention-directing vs. neutral cues, despite the robust spatial bias. Our data thus provide clear "proof of principle" that spatial biases in microsaccades *can* exist without the addition of new microsaccades. Furthermore, we show that this is the case even when ongoing microsaccade rate is low preceding the cue, suggesting that our findings are not restricted to situations where microsaccade rate is already close to ceiling. At the same time, we cannot rule out that there may be other settings in which attention shifts could directly trigger microsaccades, such as when there are no concurrent demands on fixation (i.e., in the absence of a fixation marker). Assessing this possibility remains an interesting target for future research.

## Conclusion
In conclusion, we report a clear dissociation between the influence of spatial attention shifts on microsaccade direction vs. microsaccade rate. We observed a robust biasing of microsaccade direction by the direction of internal attention shifts, without a corresponding change in overall microsaccade rate. This shows that the spatial microsaccade modulation must be attributed to a directional biasing of ongoing microsaccades rather than to the addition of new microsaccades that are triggered by the spatial shift of attention itself. This helps to explain the probabilistic nature of the link between microsaccades and spatial attention and provides relevant context for future work delineating the links between spatial attention shifts, microsaccades, and upstream oculomotor brain circuitry.

## Data availability
The raw behavioural and eye data is publicly available at: https://osf.io/r4dgk/.

## Code availability
Analysis code (in Matlab) used for this study is publicly available at: https://osf.io/r4dgk/. See also https://osf.io/w8c4b/ for our saccade-detection functions (Matlab code) and an elementary version of our experimental task (Python code).

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

## Acknowledgements

This work was supported by an ERC Starting Grant from the European Research Council (MEMTICIPATION, 850636) and a NWO Vidi Grant from the Dutch Research Council (14721) to F.v.E. The funder had no role in study design, data collection and analysis, decision to publish or preparation of the manuscript. The authors also wish to thank Chris Jungerius and Caterina Trentin for their input during the revision of this article.

## Author contributions

B.L. and F.v.E. designed the study. B.L. programed the experiment. Z.-S.A. conducted the data collection; B.L. and F.v.E. conducted the analysis. B.L. and F.v.E. wrote and revised the manuscript.

## Competing interests

The authors declare no competing interests.
