## [Transparent Peer Review file · Communications Psychology]

Attentional shifts bias microsaccade direction but do not cause new microsaccades

Corresponding Author: Mr Baiwei Liu

Version 0:

Decision Letter:

Dear Mr Liu,

Thank you for your patience during the peer-review process. Your manuscript titled "Attention and microsaccades: why is the link between microsaccades and attention shifts probabilistic?" has now been seen by 3 reviewers (numbered #2,#3,#4), whose comments are appended below. You will see that they find your work of some potential interest. However, they have raised quite substantial concerns that must be addressed. In light of these comments, we cannot accept the manuscript for publication, but would be interested in considering a revised version that fully addresses these serious concerns.

We hope you will find the Reviewers' comments useful as you decide how to proceed. Should additional work allow you to address these criticisms, we would be happy to look at a substantially revised manuscript. If you choose to take up this option, please highlight all changes in the manuscript text file, and provide a detailed point-by-point reply to the reviewers.

The Reviewers highlight in particular that the work, in its present form, does not present a sufficiently strong advance for specialists in the field. The revision therefore must include additional empirical work to address potential confounds and provide clearer insights into the mechanisms.

Editorially, we consider the following to be the most crucial aspects of a revision:

1. Provide evidence, from a new experiment or analysis, that microsaccade rate is not effectively at ceiling while fixating on the cue (suggested by Reviewer 2 as a potential explanation for the lack of increase in microsaccades).
2. Provide evidence, from a new experiment or analysis, that dissociates general attentional focusing from spatial orienting, and considers the research question (i.e. "are new microsaccades generated?") separately for each (as suggested by Reviewer 3).
3. Explain more thoroughly the advance offered by the research, considering previous studies that have analysed microsaccade rate changes during spatial attention shifts (e.g. Laubrock et al 2005, Engbert & Kliegl 2003, Laubrock et al 2010). Please note that what we are asking for is a clear delineation of how your work confirms, contradicts, or extends previous work. We consider a strong confirmation a potential advance, and strongly advise against unjustified novelty claims or representing the existing literature as more divisive or incomplete than is the case.

I am attaching a checklist that details critical reporting requirements for the revised manuscript. Please attend to each item and ensure your manuscript is fully compliant. We are requesting that your manuscript aligns with these requirements as this facilitates the evaluation of your manuscript, reducing delays in re-review and potential future acceptance. If your revised manuscript is not aligned with these requests on major issues, such as those concerning statistics, it may be returned to you for further revisions without re-review. Additional information can be found in our style and formatting guide Communications Psychology formatting guide.

If the revision process takes significantly longer than five months, we will be happy to reconsider your paper at a later date, provided it still presents a significant contribution to the literature at that stage.

Please use the following link to submit your

- revised manuscript,
- point-by-point response to the referees' comments,
- cover letter (as a separate document),
- the Editorial Policy Checklist (see below),
- the Reporting Summary (see below), and
- the completed Editorial Request Table (attached):

Link Redacted

Thank you for the opportunity to review your work.

Best regards,

Kate Storrs

Katherine Storrs, PhD
Editorial Board Member
Communications Psychology
orcid.org/0000-0001-9573-8654

REVIEWER EXPERTISE:

Reviewer #2: visual attention + eye movements

Reviewer #3: eye movements + eye tracking

Reviewer #4: visual attention + eye movements + eye tracking

REVIEWER REPORTS:

Reviewer #2 (Remarks to the Author):

This is a very elegant study that shows a strong dissociation between modulation in microsaccades' direction versus microsaccades' rate. I think the findings are very important for the field and certainly deserve a publication. I have only minor suggestions.

Figure 3. I think it would be helpful to replicate Figure3AB but for neutral condition. I would expect to see no modulation but that would 1) confirm expectations of the reader about study design, 2) additionally address a worry about the "null result" by showing a clear difference between the two conditions with respect to directional selectivity. This is already implied but I think extra visualization will be helpful.

Figure 2. (A) Logistic/binomial equivalent of a t-test would be more suitable than a paired t-test. (B) given the skewed distribution of RTs, it would make sense to use geometric means for plotting and a LMM on log-transformed RTs for analysis. I don't have much experience working with Jamovi, but I think it has both analyses built in. I do not expect this to change the results, but be more suitable from not-violating LM assumptions point of view.

Reviewer #3 (Remarks to the Author):

The present study further investigates the link between microsaccades and selection in visual working memory. More specifically, the authors asked whether shifting attention in memory can (obligatorily) trigger unique microsaccades, or whether the link between attention and microsaccades in a memory task is more probabilistic. The probabilistic account predicts that attention shifts do not elicit new microsaccades, but only bias the direction of ongoing microsaccades. The authors present one experiment using the same paradigm that this group has used in a number of very insightful studies over the past few years. The main idea of that protocol is to use a retro-cue that itself does not require any directional bias in

microsaccades, but participants still show a directional bias as a function of the spatial position of the selected memory item. In this study, they compared microsaccade rates in trials with neutral vs valid retro-cues and found that the frequency of microsaccades was indistinguishable between the two conditions, while valid retro-cues still resulted in a directional bias. The authors conclude that attentional orienting in memory does not elicit additional microsaccades, therefore supporting a probabilistic link between microsaccades and attention shifts. They provide a conceptual normalization model that can account for the findings.

This is a very well written manuscript using an established experimental paradigm. Both the data obtained and the analyses are technically sound and convincing. Nevertheless, the manuscript does little to advance our understanding of the relationship between attention and microsaccades. The topic has been studied extensively over the last two decades, and the new study presented here uses standard approaches to mainly confirm previous conclusions. I will describe my concerns in more detail below.

Concerns:

1. The question of whether shifts of attention elicit new additional microsaccades has been investigated in the perceptual domain, which the authors also acknowledge. However, the authors do not motivate why the results should be different when this relationship is investigated in a visual working memory task. The authors argue at one point that their paradigm has the advantage of not presenting a visual stimulus that could lead to a biased response. Is this the only motivation for revisiting this research question, or is there also a theoretical motivation for studying this link in the memory domain?
2. Microsaccades are not some kind of epiphenomenal brain signature. They have a function, and that function is primarily to keep the eye on a fixation point. In this sense, it is not surprising that valid retro-cues elicit the same microsaccade rate as neutral retro-cues. It would be important to know whether the relationship between attention and microsaccades varies with the overall microsaccade rate. For example, the oculomotor system may already be working at its peak microsaccade rate to keep the eyes fixated on the fixation symbol. Removal of the fixation symbol typically results in fewer microsaccades because there is no reference point to correct the current gaze. It would be important to know whether new microsaccades can be elicited in this situation, which would strongly influence the authors' main conclusion.
3. While I appreciate that the authors are trying to propose a qualitative mechanism, I think the authors should implement this link in a computational model to advance the field. The normalisation model has been implemented many times as a computational model for various tasks. Similarly, there is already a computational model that has investigated microsaccade rates and orientations in response to attentional cues (Engbert, 2012). An important advance for the field would now be to quantitatively compare an existing model with a new one.

Reviewer #4 (Remarks to the Author):

The paper describes a set of related analyses of the effects cued internal attention shifts to memorized locations on microsaccade rate and directional biases. The research question was whether internal attention shifts just bias microsaccade direction (as they have shown before), or add new microsaccades. Results clearly show directional biases with little evidence or rate modulation; if anything attention shifts temporarily reduced microsaccade rate, but that reduction was not significant.

The short paper is written in a clear style and easy to follow. The data analyses appear sound. I have only a few comments.

The similarity to the autonomous saccade timing assumption in the SWIFT model of eye movement control should be discussed. Autonomous saccade timing is a fundamental assumption of SWIFT since its first instantiation (Engbert, Longtin & Kliegl, 2002). The model assumes that cognitive processing can inhibit the timer, but it does not assume that cognitive processing generated additional saccades. The present manuscript similarly assumes autonomous microsaccade timing, and provides evidence against the generation of additional microsaccades by voluntary attention shifts. In fact, I wonder what motivated the assumption that additional microsaccades might be inserted, to me it reads like something of a straw man (but I might be willing to tolerate that because it makes the story more interesting).

Attentional focusing and spatial attention shifts should be told apart. Currently Microsaccade rates have been shown to be modulated by external and internal attention. Several such effects are well documented. First, microsaccade rate drops before cue onset when constant ISIs are used. Second, there is a well-replicated modulation of microsaccades following cue onsets, with an initial drop followed by a later period of enhanced microsaccade rate (e.g., Engbert & Kliegl, 2003; Laubrock et al, 2005, 2010), and even with equiluminant cues (Rofls et al., 2008), just as if an ongoing microsaccadic plan were inhibited by a cue. This is also visible in the present data. It is not discussed as an attentional effect, although it is likely to be a signal of alertness. This possibility should at least be mentioned. Third, microsaccades are influenced by perceptual switches in ambiguous displays (Laubrock, Engbert & Kliegl, 2008, Hsieh & Tse, 2009) and hence indicate changes in perceptual awareness, which they sometimes even cause (Martinez-Conde et al., 2006). I would consider all of these related to attention, but not necessarily to spatial attention. Your results suggest to me that the cue first triggers alertness-induced attentional focusing, which is then followed with some delay by a voluntary shift of spatial attention to the memorized location indicated by the cue. The authors should disentangle several attentional effects, and make clear that their conclusion that internal attention shifts bias direction, but do not affect microsaccade rate holds for spatial attention shifts, but not the alertness signature. I would even be more cautious and leave open the possibility that internal shifts might cause an

additional weak inhibition of microsaccades which is certainly rather weak if it exists at all (i.e., the blue-colored dots in Fig 4b panel 3 and the numerically higher rates in Figure 3c, see also your comment on ll. 360), and might be detectable with more power.

minor comments:

- I. 1 The "why" in the title is not really answered, maybe remove it?

- Figure 4: The color scales are confusing. It doesn't make sense to use the same scale for the upper four as for the lower two panels, because they report different quantities (absolute rates vs. rate differences). I suggest to change the color scale for the upper four panels to only include positive values.

- Figure 4: consider using a logarithmic y scale, ideally with labels still from 1 to 5, to enlarge the part of the figure in which the action happens (there's not much going on beyond 2 degrees, but of course the axis needs to be plotted up to 5 degrees because of the target locations).

- II. 483ff the ideas for future studies seems a bit random, so let me add one: it would be interesting to see how the size effect scales with presence vs. absence of a fixation symbol. Would the results be more Spivey-like (l. 238) if a cue like background color or outer frame color were used?

EDITORIAL POLICIES

We ask that you ensure your manuscript complies with our editorial policies and reporting requirements.

To that end, we require revised manuscripts to be accompanied by two completed items: a reporting summary that collects information on study design and procedure, and an editorial policy checklist that verifies compliance with all required editorial policies

- <https://www.nature.com/documents/nr-reporting-summary.zip>>Nature Research Reporting Summary
- <https://www.nature.com/documents/nr-editorial-policy-checklist.pdf>>Editorial Policy Checklist

All points on the policy checklist must be addressed. Your revised manuscript can only be sent back to the referees if these checklists are completed and uploaded with the revision.

Notes: If you have submitted a Stage 1 Registered Report, Review, Primer, Comment, or Perspective you do not need to submit these forms. If you have already submitted these forms, you may disregard this request.

Version 1:

Decision Letter:

Dear Mr Liu,

Thank you for your patience during the peer-review process. Your manuscript titled "Attention and microsaccades: why is the link between microsaccades and spatial attention shifts probabilistic?" has now been seen again by the original 3 reviewers (numbered #2,#3,#4 as perviously), and I include their comments at the end of this message. As you will see, the reviewers appreciate the improvements made to the manuscript and are positive about the work. We are interested in the possibility of publishing your study in Communications Psychology, but would like you to incorporate the remaining statistical suggestions that arose during review.

We therefore invite you to revise and resubmit your manuscript, along with a point-by-point response to the reviewers. Please highlight all changes in the manuscript text file. In line with the Communications Psychology guidelines on statistical reporting, we request that you check the normality of the data before reporting Cohen's D, and if assumptions are not met, report an alternative non-parametric effect size estimate. Likewise, we ask that you implement the Linear Mixed Model advised by the reviewer; analyses should be guided by statistical principle rather than status quo where innovation presents an improvement.

I am attaching an Editorial Requests Table that details critical reporting requirements for the revised manuscript. Please attend to each item and ensure your manuscript is fully compliant. We are requesting that your manuscript aligns with these requirements as this facilitates the evaluation of your manuscript, reducing delays in re-review and potential future acceptance. If your revised manuscript is not aligned with these requests on major issues, such as those concerning statistics, it may be returned to you for further revisions without re-review. Additional information can be found in our style and formatting guide <https://www.nature.com/documents/commspsychol-style-formatting-guide-accept.pdf> Communications Psychology formatting guide.

Please use the following link to submit your

- revised manuscript,
- point-by-point response to the referees' comments,
- cover letter (as a separate document),
- the Editorial Policy Checklist (see below),
- the Reporting Summary (see below), and
- the completed Editorial Request Table (attached):

Link Redacted

Best regards,

Kate Storrs

Katherine Storrs, PhD
Editorial Board Member
Communications Psychology
orcid.org/0000-0001-9573-8654

REVIEWER EXPERTISE:

Reviewer #2 visual attention + eye movements

Reviewer #3 eye movements + eye tracking

Reviewer #4 visual attention + eye movements + eye tracking

REVIEWER REPORTS:

Reviewer #2 (Remarks to the Author):

The authors' response on statistical analysis makes me very sad. One cannot but wonder how many more dozens(!) of years must pass until linear mixed models stop being thought of "sophisticated" and "complex". This is a very minor but important advancement over unregularized repeated measurements test such paired t-test or rm ANOVA (LMM tend to overfit less). The difference is literally in regularized (finite variance, LMM) versus unregularized (infinite variance flat priors, LM) distributions for random effects. Same goes for generalized linear models, is a century enough for logistic regression to finally become widely used on binomial data without converting to proportions and forcing it into an ill-suited linear model analysis such a t-test or ANOVA. throwing out all information about uncertainty in the process? It is also odd to use Cohen's D that works ONLY for normal distributions without checking for that. Cohen's D is effectively a measure of overlap between two normal(!) distributions but if distributions are not normal the formula no longer matches the overlap and is misleading. I understand the allure of linear models, but it only works if your data allows for it, not when you wish it would.

Science relies on the use of appropriate statistics that match outcome variables. Not traditional, not mainstream, not convenient for comparison across different measures of different nature, but one that matches outcome variable distribution and assumptions about generative process. Repeating ill-suited analysis only propagates it further into mainstream (this is literally "if everyone jumped off the bridge" situation).

Reviewer #3 (Remarks to the Author):

The authors answered all my concerns very thoroughly and in great detail. In particular, I appreciate the additional analysis that the authors have provided. I have no further concerns.

Reviewer #4 (Remarks to the Author):

I thank the authors for addressing all of my points to my satisfaction, and congratulate them on a fine paper.

EDITORIAL POLICIES

We ask that you ensure your manuscript complies with our editorial policies and reporting requirements.

To that end, we require revised manuscripts to be accompanied by two completed items: a reporting summary that collects information on study design and procedure, and an editorial policy checklist that verifies compliance with all required editorial policies.

- <https://www.nature.com/documents/nr-reporting-summary.zip>>Nature Research Reporting Summary
- <https://www.nature.com/documents/nr-editorial-policy-checklist.pdf>>Editorial Policy Checklist

All points on the policy checklist must be addressed. Your revised manuscript can only be sent back to the referees if these checklists are completed and uploaded with the revision.

Notes: If you have submitted a Stage 1 Registered Report, Review, Primer, Comment, or Perspective you do not need to submit these forms. If you have already submitted these forms, you may disregard this request.

Version 2:

Decision Letter:

Dear Mr Liu,

Your manuscript titled "Attention and microsaccades: why is the link between microsaccades and spatial attention shifts probabilistic?" has now been seen by our reviewer, whose comments appear below. In light of their advice I am delighted to say that we are happy, in principle, to publish a suitably revised version in Communications Psychology.

We therefore invite you to revise your paper one last time to address a list of editorial requests. At the same time we ask that you edit your manuscript to comply with our format requirements and to maximise the accessibility and therefore the impact of your work.

EDITORIAL REQUESTS:

SUBMISSION INFORMATION:

OPEN ACCESS:

* **CODE AVAILABILITY:** All Communications Psychology manuscripts must include a section titled "Code Availability" at the end of the methods section. We require that the custom analysis code supporting your conclusions is made available in a

publicly accessible repository at this stage; please choose a repository that generates a digital object identifier (DOI) for the code; the link to the repository and the DOI must be included in the Code Availability statement. Publication as Supplementary Information will not suffice.

*** DATA AVAILABILITY:**

Link Redacted

Best regards,

Marike

on behalf of Katherine Storrs

Marike Schiffer, PhD
Chief Editor
Communications Psychology

REVIEWERS' COMMENTS:

Reviewer #2 (Remarks to the Author):

Thank you for going more thorough!

Dear Reviewers,

Thank you for having taken the time to carefully evaluate our work, and for each of your valuable comments and suggestions for improvement.

Judging by the reviewers' summary comments, we were pleased to read that all three reviewers (R2, R3, R4) considered our write-up clear and our findings sounds and convincing. For example:

- R2: *“This is a very elegant study [...]. I think the findings are very important for the field and certainly deserve a publication. I have only minor suggestions.”*
- R3: *“This is a very well written manuscript [...]. Both the data obtained and the analyses are technically sound and convincing.”*
- R4: *“The short paper is written in a clear style and easy to follow. The data analyses appear sound. I have only a few comments.”*

Besides this overall appreciation of our work, we were of course also pleased with the constructive expert-level comments and suggestions that we received – and that we have embraced to improve our manuscript. In response to the fair and thoughtful comments of all three reviewers, we have (1) included the outcomes of multiple additional analyses, including relevant control analyses, that further strengthen our conclusions, (2) tightened our terminology to make clear that our findings refer specifically to “spatial attention”, and (3) added key clarifications and discussion of related literature at relevant instances throughout our article.

Please find our point-by-point replies to each of the reviewer comments below.

We are grateful for your valuable time and thoughts in (re)evaluating our work.

Sincerely,

Baiwei Liu, Zampeta-Sofia Alexopoulou, Freek van Ede

Point-by-point replies to reviewers

Reviewer 1

N/A. (i.e., we did not receive comments from “reviewer 1”).

Reviewer 2

This is a very elegant study that shows a strong dissociation between modulation in microsaccades' direction versus microsaccades' rate. I think the findings are very important for the field and certainly deserve a publication. I have only minor suggestions.

Thank you – we are of course very pleased to read such a positive appreciation of our study and findings. Thank you also for your valuable suggestions that we have embraced to further improve the quality of our work. We finally wish to thank you for providing us with this neat way to summarise our main results, that we have now adopted throughout our article. For example:

Page 9, Results:

This dissociation whereby spatial shifts of attention modulate saccade direction (**Fig. 3c**) but not rate (**Fig. 3d**) implies that spatial-attention shifts do not generate new microsaccades.

Page 14, Discussion:

In conclusion, we report a clear dissociation between the influence of spatial attention shifts on microsaccade direction vs. microsaccade rate. [...]

Figure 3. I think it would be helpful to replicate Figure3AB but for neutral condition. I would expect to see no modulation but that would 1) confirm expectations of the reader about study design, 2) additionally address a worry about the "null result" by showing a clear difference between the two conditions with respect to directional selectivity. This is already implied but I think extra visualization will be helpful.

Thank you for this great suggestion. We agree that this provides an excellent way to showcase our key dissociation between attentional effects on microsaccade *direction vs. rate*, and to better appreciate the logic of our experimental design. In accordance with this great suggestion, we have now revised our key figure to also include the comparison of the spatial bias following attention-directing and neutral cues, and have adjusted the relevant results description accordingly:

Page 7-8, Results:

[...] For the purpose of enabling a direct comparison between attention-directing and neutral cues, we also considered saccade rates following neutral cues (**Fig. 3b**) whose colour did not match either memory item and therefore did not invite a shift of spatial attention to either memory item. For this, we artificially classified saccades as “toward” or “away” based on the memory item that would *eventually* be tested – but that could not yet be known in the period of interest after the cue. Because neutral cues did not inform which item would become tested, it was re-assuring that after neutral cues we observed no difference in the number of “toward” and “away” saccades (**Fig. 3b**, bar graph; $t(23) = -0.285$, $p = 0.778$, $d = -0.058$). **Figure 3c** shows the overlay of the spatial biasing of microsaccades (toward minus away) following cues that did vs. did not invite a spatial shift of attention to either memory item. A direct comparison showed significantly larger spatial biasing following attention-directing vs. neutral cues, both when comparing the full time courses (cluster $P = 0.005$ and 0.027) or the spatial bias averaged over the a-priori defined window of interest (**Fig. 3c**, bar graph; $t(23) = 3.733$, $p = 0.001$, $d = 0.762$). [...]

(Updated) Figure 3. Internal selective attention modulates the direction of microsaccades without changing overall microsaccade rate. **a)** Saccade rates after cue onset in trials with attention-directing cues for saccades in the direction of the memorised location of the cued memory item (toward) and in the opposite direction (away). **b)** Saccade rates after cue onset in trials with neutral cues. Note how neutral cues do not inform which item will become tested later on, rendering no meaningful difference between “toward” and “away” in this post-cue interval of interest (toward and away we here defined based on which item would later be tested, and serve only to showcase as a reference and to confirm the absence of any spatial bias following neutral cues). **c)** Spatial saccade-direction bias after cue onset following attention-directing and neutral cues. **d)** Overall saccade rates after cue onset following attention-directing and neutral cues. Note how saccade rates in panel d are higher than in panels a-b, given that overall saccade rates include both toward and away saccades. Horizontal lines denote significant difference clusters following cluster-based permutation (Maris and Oostenveld, 2007). Bar graphs show data averaged over the a-priori defined window from 200-600 ms after cue onset (based on (Liu et al., 2022)). Error bars indicate ± 1 SEM calculated across participants ($n=24$). Grey lines denote individual participants.

Figure 2. (A) Logistic/binomial equivalent of a t-test would be more suitable than a paired t-test. (B) given the skewed distribution of RTs, it would make sense to use geometric means for plotting and a LMM on log-transformed RTs for analysis. I don't have much experience working with Jamovi, but I think it has both analyses built in. I do not expect this to change the results, but be more suitable from not-violating LM assumptions point of view.

Thank you for this suggestion. We used paired-samples t-tests as a mainstream way to compare trial-average scores between two conditions in a within-participant design. This also enabled us to adopt the same approach for all relevant comparisons (accuracy, RT, microsaccade rate). Please note that we did not factor in single-trial data distributions (that for accuracy indeed are binary, and for RT are skewed), but rather used trial averages that we tested for consistency at the group level across participants. Unlike the single-trial measurements themselves, these trial averages tend to approximate a normal distribution at the group level.

Having clarified this, we agree that trial-average “mean RT” is suboptimal because of the skewed nature of RT distributions. Because we prefer to keep things simple and consistent between analyses, we now plot and compare *median* RTs that are more robust to the skewed nature of RT distributions. Specifically, for each participant and condition we now plot the across-trial-median RT and compare these median-RT scores between conditions at the group level using paired sample t-tests. In our view this strikes a good balance between (1) being mindful of the skewed nature of RT data, while (2) keeping things simple by adopting consistent statistical analyses throughout the article.

We also wish to acknowledge that we ourselves are not highly familiar with the more sophisticated Linear-Mixed-Modelling approach to statistics. While we do not in any way doubt the power and merit of such more complex analyses, for the current purpose, we prefer to refrain from using analyses that we are not sufficiently familiar with. In this regard, please also note how these specific accuracy and RT

comparisons primarily serve to confirm a established effect. Our central conclusions and novel insight regarding a dissociation between attentional modulations of microsaccade direction vs. rate do not hinge on these specific RT and accuracy comparisons.

Reviewer 3

The present study further investigates the link between microsaccades and selection in visual working memory. More specifically, the authors asked whether shifting attention in memory can (obligatorily) trigger unique microsaccades, or whether the link between attention and microsaccades in a memory task is more probabilistic. The probabilistic account predicts that attention shifts do not elicit new microsaccades, but only bias the direction of ongoing microsaccades. The authors present one experiment using the same paradigm that this group has used in a number of very insightful studies over the past few years. The main idea of that protocol is to use a retro-cue that itself does not require any directional bias in microsaccades, but participants still show a directional bias as a function of the spatial position of the selected memory item. In this study, they compared microsaccade rates in trials with neutral vs valid retro-cues and found that the frequency of microsaccades was indistinguishable between the two conditions, while valid retro-cues still resulted in a directional bias. The authors conclude that attentional orienting in memory does not elicit additional microsaccades, therefore supporting a probabilistic link between microsaccades and attention shifts. They provide a conceptual normalization model that can account for the findings.

This is a very well written manuscript using an established experimental paradigm. Both the data obtained and the analyses are technically sound and convincing.

Thank you for your careful evaluation of our work and for considering our analyses and findings sound and convincing. Thank you also for your valuable suggestions that have prompted us to run additional analyses as we turn to below. The outcomes of these additional analyses reinforce our conclusions and provide another relevant advance over related prior work.

Nevertheless, the manuscript does little to advance our understanding of the relationship between attention and microsaccades. The topic has been studied extensively over the last two decades, and the new study presented here uses standard approaches to mainly confirm previous conclusions. I will describe my concerns in more detail below.

Concerns:

1. The question of whether shifts of attention elicit new additional microsaccades has been investigated in the perceptual domain, which the authors also acknowledge. However, the authors do not motivate why the results should be different when this relationship is investigated in a visual working memory task. The authors argue at one point that their paradigm has the advantage of not presenting a visual stimulus that could lead to a biased response. Is this the only motivation for revisiting this research question, or is there also a theoretical motivation for studying this link in the memory domain?

This is an important point that we have been mindful of ever since we decided to pursue our study, as well as when writing up our findings. In short, our findings do not contradict prior work, but extend it in important ways – as we spell out, and have now expanded upon, at relevant instances in our introduction and discussion. We also re-iterate these key points below. In addition, please note how the new results we now added in response to point 2 below (showing that our lack of an increase in overall rate is not due to a “ceiling effect”), provides yet another novel and relevant advance.

First, while we explicitly acknowledge that the comparison between overall microsaccade rates following attention-directing vs. neutral cues was also available in prior studies, we also point out how in several of these key early studies the cue itself was not carefully matched between attention-directing vs. neutral cues (e.g. different colours, without counterbalancing). This makes it notoriously hard to isolate the influence of top-down goal-directed attention from bottom-up cue processing. Indeed, it has been shown how bottom-up cue features can also modulate microsaccades (Meyberg et al., 2017b, 2017a), thus contaminating interpretation if not carefully matched between informative and neutral cues. In our study, we used non-spatial colour-cues that were carefully matched and counterbalanced between informative (attention-directing) and neutral cues, to yield a clean interpretation necessary for addressing our question.

Second, in prior studies, spatial modulations in microsaccades were relatively weak, in comparison to the spatial modulations we have reported in recent years in the context of our working-memory paradigm (where there is no external target to “try to look at”). Importantly, having a profound spatial modulation in the first place is a clear prerequisite for studying our central question as to what “accounts” for this spatial modulation.

Third, our key question and key data comparisons were never the “central point” in any of these prior studies (at least not in our reading of this work). By putting our central question up front, we address a relevant point that brings together a number of contemporary findings (such as why microsaccades correlate with neural signatures of spatial attention, but are not necessary for such modulations to occur).

In addition to above clarifications, another advance is that our findings were made within the context of internal attention shifts to the contents of working memory (rather than external attention shifts to the sensory contents of perception). This is important in light of a number of related studies that have started to investigate similar biases in the context of working memory, including not only studies from our lab but also recent studies from other labs (that we now also cite). Our current findings help to better understand and interpret such microsaccade biases in working memory, that we anticipate will continue to be used as a marker of internal attention shifts in the future.

We bring these important points forward at key dedicated places in our abstract, introduction, and discussion:

Page 1, Abstract:

We utilised an internal-selective-attention task that has recently been shown to yield robust spatial microsaccade modulations and compared microsaccade rates following colour retrocues that were carefully matched for bottom-up sensory input, but differed in whether they invited an attention shift or not.

Page 2-3, Introduction:

We note how the comparison between informative (attention-directing) and neutral cues was also available in (Engbert and Kliegl, 2003; Laubrock et al., 2005) though in these studies the spatial biasing by voluntary shifts of spatial attention itself was weak and the cues were not perfectly matched, making it hard to address the question that we put central here. Indeed, it has been shown how bottom-up cue features can also modulate microsaccades (Meyberg et al., 2017b, 2017a), thus contaminating interpretation if not carefully matched between informative and neutral cues. Here we carefully matched informative and neutral cues and studied microsaccade-rate modulations in the context of an internal selective-attention task that we recently established as a powerful model system because it yields robust spatial modulations in microsaccades by spatial shifts of attention (here, directed within the spatial layout of visual working memory) (van Ede et al., 2019, 2020; Liu et al., 2022, 2023; de Vries et al., 2023; Wang and van Ede, 2024).

Page 13-14, Discussion:

At least two early studies on microsaccade biases by visual-spatial attention also included neutral cues (Engbert and Kliegl, 2003; Laubrock et al., 2005). While their data thus allowed the same key comparison as we targeted here, there are several relevant differences. One relevant difference is that in these studies the employed endogenous cues showed only weak directional biasing of microsaccades (Laubrock et al., 2005). In comparison, here, we observed a clear bias; building on our prior work using the same overall task set-up (e.g., (van Ede et al., 2019, 2020, 2021; Liu et al., 2022, 2023; de Vries et al., 2023; Wang and van Ede, 2024)). It was only in the presence of this clear spatial modulation that our question was of interest: whether we could “explain” this spatial modulation by the addition of new microsaccades. Second, in aforementioned work, the authors did not fully match informative (attention-directing) and neutral cues in terms of bottom up cue features. This complicates a direct comparison given that differences in bottom-up cue features can also lead to differences in microsaccades (Meyberg et al., 2017b, 2017a). In contrast, we always used the same colour cues, and counterbalanced whether specific colours were attention-directing or neutral. Third, we here included a relevant additional analyses to show that the lack of an overall rate increase is not due to a ceiling effect. Finally, we uniquely compared microsaccades following attention-directing versus neutral cues in the context of an internal selective attention task in which spatial attention was directed internally

to the contents of working memory – thus extending prior observations from the domain of perception to the domain of working memory.

In addition, please note how our the outcomes from our additional analysis in response to point 2 below provide yet another relevant advance over prior work (in addition to: (1) using informative and neutral cues that were carefully matched and counterbalanced, (2) starting from a profound spatial modulation that is a pre-requisite for addressing our central question, and (3) studying our question in the context of working memory).

2. Microsaccades are not some kind of epiphenomenal brain signature. They have a function, and that function is primarily to keep the eye on a fixation point. In this sense, it is not surprising that valid retro-cues elicit the same microsaccade rate as neutral retro-cues. It would be important to know whether the relationship between attention and microsaccades varies with the overall microsaccade rate. For example, the oculomotor system may already be working at its peak microsaccade rate to keep the eyes fixated on the fixation symbol. Removal of the fixation symbol typically results in fewer microsaccades because there is no reference point to correct the current gaze. It would be important to know whether new microsaccades can be elicited in this situation, which would strongly influence the authors' main conclusion.

Thank you for this relevant reflection. We agree that this point about a potential “ceiling effect” is fair and relevant, and it has prompted us to include the outcomes of an additional control analysis that directly addresses this point.

Before turning to this control analysis, we first wish to clarify that even if microsaccade rates were at ceiling, our data still provide “proof-of-principle” evidence for our central conclusion that spatial modulations in microsaccades *can* exist without the addition of new microsaccade (indeed, reasoning from our scenario in which attention “triggers” additional microsaccades, we should have observed no attentional modulation if microsaccade rate was already “at ceiling”; yet, we found a clear modulation).

Having clarified this, we agree that this does not imply that attention may never trigger new microsaccades. Indeed, this point raises the fair question whether there may *also* be instances where attention *will* drive new microsaccades, such as when ongoing microsaccade rate is low, and thus not close to a potential “ceiling”. To directly address this possibility in our own data, we have performed a key additional analysis where we sorted trials by their ongoing microsaccade rate prior to the cue, generating “high-rate trials” (closer to a potential ceiling) and “low-rate trials” (further from a potential ceiling). As we show below, our key findings were preserved in both cases – showing a clear spatial bias in microsaccade direction without an overall increase in rate, regardless of how “close-to-ceiling” the ongoing microsaccade rate was preceding cue onset. Because we believe these findings provide important additional insight, we have now included these additional findings in a new section and an additional figure that we have added to our results:

Page 9, Results:

The lack of an overall increase in microsaccade rate is not due to a ceiling effect

We report a clear spatial modulation in microsaccade direction (**Fig. 3c**) without a concomitant increase in overall microsaccade rate (**Fig. 3d**). This makes clear that spatial modulations in microsaccades *can* exist without the addition of new (spatial-attention-triggered) microsaccades. At the same time, it remains possible that our findings are specific to cases where microsaccade rates are already close to ceiling, such that no new microsaccades can be added. To address this possibility, we sorted our trials by the rate of saccades in the second preceding the cue (using a median split), yielding “high-rate trials” (where ongoing saccade rate was closer to a putative ceiling) and “low-rate trials” (where ongoing saccade rate was further from ceiling). If the lack of an overall increase in the number of microsaccade following attention-directing vs. neutral cues is due to a ceiling effect, we should find this lack of an overall increase in rate only in the high-rate, but not the low-rate trials. In contrast, we observed the same qualitative pattern of results after the cue regardless of preceding saccade rate (**Fig. 4**). While pre-cue saccade rates confirmed our trial sorting, we found that high-rate (**Fig. 4a,c**) and low-rate (**Fig. 4b,d**) trials each showed a clear spatial modulation following attention-directing cues (**Fig. 4a,b**; cluster $P = 0.032$, 0.0001 ; bar-graph comparisons: $t(23) = 3.302$, $p = 0.031$, $d = 0.674$; $t(23) = 4.564$, $p = 0.0001$, $d = 0.932$ for high-rate and low-rate trials respectively) without an increase in overall saccade rate (**Fig. 4c,d** no clusters found; bar-graph comparisons: $t(23) = -1.699$, $p = 0.103$, $d = -0.347$; $t(23)$

= -1.902, $p = 0.078$, $d = -0.388$). This shows that our key finding – a lack of an overall increase in rate, despite a clear spatial modulation – is not restricted to cases where saccade rate is already close to ceiling.

(New) Figure 4. The lack of an overall rate increase is not due to a ceiling effect. **a)** Toward and away saccades following attention-directing cues in trials with high saccade rates preceding cue onset (median split). **b)** Toward and away saccades following attention-directing cues in trials with low saccade rates preceding cue onset. **c)** Overall saccade rates after cue onset following attention-directing and neutral cues in trials with high saccade rates preceding cue onset. **d)** Overall saccade rates after cue onset following attention-directing and neutral cues in trials with low saccade rates preceding cue onset. Conventions as in Figure 3.

In addition, we now also bring back this relevant point – and these relevant new analyses – in our Discussion:

Page 14, Discussion:

“[...] Our data thus provide clear “proof of principle” that spatial biases in microsaccades can exist without the addition of new microsaccades. Furthermore, we show that this is the case even when ongoing microsaccade rate is low preceding the cue, suggesting that our findings are not restricted to situations where microsaccade rate is already close to ceiling. At the same time, we cannot rule out that there may be other settings in which attention shifts could directly trigger microsaccades, such as when there are no concurrent demands on fixation (i.e., in the absence of a fixation marker). Assessing this possibility remains an interesting target for future research.”

In addition, please note how even if the primary purpose of microsaccades in our task is to re-fixate, microsaccades nevertheless track internal attention shifts over and above the re-fixation goal. Our data show that this happens without adding new microsaccades. This provides direct evidence for our central conclusion for the “proof of principle” that spatial modulations in microsaccades *can* exist without the addition of new microsaccades. Whether there may be other cases in which attention *can* directly trigger new microsaccades is an interesting question that we now acknowledge (see added text above).

3. While I appreciate that the authors are trying to propose a qualitative mechanism, I think the authors should implement this link in a computational model to advance the field. The normalisation model has been implemented many times as a computational model for various tasks. Similarly, there is already a computational model that has investigated microsaccade rates and orientations in response to attentional cues (Engbert, 2012). An important advance for the field would now be to quantitatively compare an existing model with a new one.

Thank you for this suggestion. Our study mainly served to showcase the dissociation between attentional modulations in microsaccade direction and rate, and to show this in the context of our working memory task, using carefully matched cues. We highlight the various advances over prior work

in our response to point 1 above, where we also list relevant sections in our article that articulate these advances.

Additionally developing, implementing, and comparing computation models is beyond our scope – and, admittedly, also beyond our expertise. We included a descriptive normalisation account merely for an intuitive understanding, as we motivated in our discussion:

Page 12, Discussion:

[...] We wish to make clear however that our intention here is not to prove this specific model. Rather, we merely aim to provide one possible way to make sense of our findings in which attention clearly modulates the relative rates of toward vs. away microsaccades but without a net increase in total microsaccade rate.

We hope our data inspire future researchers to further develop, integrate, and compare models, as we agree with the reviewer that this provides another important way to advance the wider field. To this end, we have now added the following to our discussion, where we also bring forward this key citation provided by the reviewer:

Page 12, Discussion:

While computational modelling of our findings is beyond the scope of the current work, it will be important in future work to integrate these ideas with existing models of microsaccade generation (such as (Engbert et al., 2002; Engbert, 2012)). In this light, it is also important to note how our findings are consistent with the autonomous saccade timing assumption in the popular SWIFT model of saccade generation (Engbert et al., 2002), which posits that cognitive processes may inhibit the timer, but not generate saccades directly.

Reviewer 4

The paper describes a set of related analyses of the effects cued internal attention shifts to memorized locations on microsaccade rate and directional biases. The research question was whether internal attention shifts just bias microsaccade direction (as they have shown before), or add new microsaccades. Results clearly show directional biases with little evidence or rate modulation; if anything attention shifts temporarily reduced microsaccade rate, but that reduction was not significant.

The short paper is written in a clear style and easy to follow. The data analyses appear sound. I have only a few comments.

Thank you for your careful evaluation of our article and for considering our article clear and easy to follow and our analyses sound. We are also grateful for your valuable input that has helped us to further strengthen our article.

The similarity to the autonomous saccade timing assumption in the SWIFT model of eye movement control should be discussed. Autonomous saccade timing is a fundamental assumption of SWIFT since its first instantiation (Engbert, Longtin & Kliegl, 2002). The model assumes that cognitive processing can inhibit the timer, but it does not assume that cognitive processing generated additional saccades. The present manuscript similarly assumes autonomous microsaccade timing, and provides evidence against the generation of additional microsaccades by voluntary attention shifts. In fact, I wonder what motivated the assumption that additional microsaccades might be inserted, to me it reads like something of a straw man (but I might be willing to tolerate that because it makes the story more interesting).

Thank you for pointing us to this excellent and highly relevant reference. It is indeed great to see that our findings are consistent with a fundamental assumption of this classic model, and we now discuss this model when interpreting our findings:

Page 12, Discussion:

[...] In this light, it is also important to note how our findings are consistent with the autonomous saccade timing assumption in the popular SWIFT model of saccade generation (Engbert et al., 2002), which posits that cognitive processes may inhibit the timer, but not generate saccades directly.

We also appreciate that this model may make the scenario that we provide evidence against (“that attention may trigger new microsaccades”) appear like a weak strawman, at least to a microsaccade expert. At the same time, when we have presented our prior microsaccade findings at scientific meetings to our usual audience of cognitive psychologists and working-memory researchers, we have come to realise that this interpretation that attention may trigger new microsaccades is commonly intuited. This is precisely the reason we set out to test this in our study. Our current findings make clear that this “common intuition” is not a valid interpretation, and it is re-assuring to know that this is not only compatible with earlier empirical work (that we already cited), but also with established saccade-generation models (as we now also cite). We are thus grateful for the reviewer for pointing us to this relevant work that reinforces our conclusion.

Attentional focusing and spatial attention shifts should be told apart. Currently Microsaccade rates have been shown to be modulated by external and internal attention. Several such effects are well documented. First, microsaccade rate drops before cue onset when constant ISIs are used. Second, there is a well-replicated modulation of microsaccades following cue onsets, with an initial drop followed by a later period of enhanced microsaccade rate (e.g., Engbert & Kliegl, 2003; Laubrock et al, 2005, 2010), and even with equiluminant cues (Rolfs et al., 2008), just as if an ongoing microsaccadic plan were inhibited by a cue. This is also visible in the present data. It is not discussed as an attentional effect, although it is likely to be a signal of alertness. This possibility should at least be mentioned. Third, microsaccades are influenced by perceptual switches in ambiguous displays (Laubrock, Engbert & Kliegl, 2008, Hsieh & Tse, 2009) and hence indicate changes in perceptual awareness, which they sometimes even cause (Martinez-Conde et al., 2006). I would consider all of these related to attention, but not necessarily to spatial attention. Your results suggest to me that the cue first triggers alertness-induced attentional focusing, which is then followed with some delay by a voluntary shift of spatial attention to the memorized location indicated by the cue. The authors should disentangle several attentional effects, and make clear that their conclusion that internal attention shifts bias direction, but do not affect microsaccade rate holds for spatial attention shifts, but not the alertness signature. I would even be more cautious and leave open the possibility that internal shifts might cause an additional weak inhibition of microsaccades which is certainly rather weak if it exists at all (i.e., the blue-colored dots in Fig 4b panel 3 and the numerically higher rates in Figure 3c, see also your comment on ll. 360), and might be detectable with more power.

Thank you for bringing this up. This comment made us realise that our occasional interchangeable use of the terms “spatial orienting” and “attentional focusing” was unintentional and undesired. We did not intend to refer, nor to make claims about, two distinct forms of attention. We had occasionally used “focusing” to refer to the *selective prioritisation* of one memory item over another. Because our memory items were lateralised at encoding, this “focusing” (in our intended use) was spatial, leading us to use “spatial attention” and “focusing” interchangeably.

This fair comment has made us realise how focusing could also be interpreted in different ways – including non-spatial attentional focusing or alertness – which was *not* our intended meaning. As we now make clear throughout our article and our updated title, our findings are specifically about spatial attention (specifically, voluntary visual-spatial shifts of attention deployed within the spatial lay-out of visual working memory) – not a generic non-spatial “focus”. Through careful experimental design, we made sure that our spatial-attention manipulation was not confounded by other, non-spatial attention processes (such as bottom-up cue processing). Addressing the link between microsaccades and other non-spatial forms of attention is beyond the scope of our article, and was never our intention. Accordingly, we no longer refer to “focus” in our article, and refer to “spatial attention” at all relevant instances. We highlight some examples below:

Page 1, Abstract:

Here, we address whether such modulations come about because spatial attention shifts trigger new microsaccades or whether, instead, spatial attention only biases [...]

Page 2, Introduction:

When considering the link between microsaccades and visual-spatial attention [...]

Page 6, Results:

Human volunteers performed a selective-attention task in which selective voluntary spatial attention was directed to one of two visual representations held within the spatial layout of working memory.

Page 7, Results:

[...] we next assessed how informative cues – that directed attention to memory items that had been presented to the left or right at encoding – modulated the direction of microsaccades, consistent with the spatial deployment of attention within the spatial layout of visual working memory.

Page 11, Discussion:

This lack of a rate increase in the face of a clear directional modulation implies that the modulation of microsaccades during visual-spatial shifts of attention [...]

In addition, we now also mention complementary work on the link between microsaccades and other, non-spatial forms, of “attentional focus”, as brought up by the reviewer:

Page 14, Discussion:

Moreover, in addition to our focus on spatial modulations in microsaccades during visual-spatial shifts of attention, it will be interesting to expand our question to non-spatial forms of attention that have also been linked to microsaccades (e.g., (Martinez-Conde et al., 2006; Laubrock et al., 2008; Hsieh and Tse, 2009)). It is further noteworthy how non-spatial forms of attention, such as temporal expectation and/or alertness, may sometimes also attenuate microsaccade rates (e.g., (Rolfs et al., 2008; Tal-Perry and Yuval-Greenberg, 2021)). Indeed, a transient increase in alertness may possibly also account for the transient drop in microsaccades (“oculomotor freezing”) that we observed after cue onset.

minor comments:

- I. 1 The "why" in the title is not really answered, maybe remove it?

Thank you for prompting us to reflect on our title. We took as a starting point the notion that the relation between microsaccades and attention *is* probabilistic. We wanted to get some handle as to “why” this is the case. For example, why are neural markers of attention shifts correlated with microsaccade when microsaccade occur, but at the same time preserved in the absence of microsaccades (as we previously reported; see Liu et al., Nature Communications, 2022)? Our data offer one key clue as to this kind of “why”, namely: *because* attention shifts themselves do not generate any microsaccades. In other words, because spatial attention shifts only modulate ongoing microsaccades, the link between spatial attention and microsaccades is “at the mercy” of there being a microsaccade in the first place. We now also make this explicit in the opening paragraph of our discussion:

Page 12, Discussion:

This pattern of results helps to understand why the link between microsaccades and attention shifts is merely probabilistic and not obligatory: because spatial attention shifts only bias ongoing microsaccades, the link between spatial attention shifts and microsaccades is “at the mercy” of there being a microsaccade in the first place.

Having clarified this, we appreciate that a “why” question can be framed at many levels, and that our data leave open other interesting why questions such as why attention does not trigger any new microsaccades. While we do not directly answer this complementary why question with our data, we also speculate about this in our Discussion when referencing the normalisation framework.

- Figure 4: The color scales are confusing. It doesn't make sense to use the same scale for the upper four as for the lower two panels, because they report different quantities (absolute rates vs. rate differences). I suggest to changed the color scale for the upper four panels to only include positive values.

Thank you for this great suggestion. We have now revised our colour maps, such that the plots showing absolute saccade rates have a colour map that only includes positive values, while the ‘effect’ plots

(showing rate *differences* between conditions) use a symmetrical colour map that runs from negative to positive. Please see our revised figure as part of our response to the comment below.

- Figure 4: consider using a logarithmic y scale, ideally with labels still from 1 to 5, to enlarge the part of the figure in which the action happens (there's not much going on beyond 2 degrees, but of course the axis needs to be plotted up to 5 degrees because of the target locations).

Thank you for this nice suggestion. The main purpose of this additional visualisation was to separate spatial biases in microsaccades from larger eye movement associated with looking back to the original item locations (“looking-at-nothing”) as has been reported in complementary studies. Accordingly, the rightful observation that there is “not much going on beyond 2 degrees” is precisely the take-home message from these plots.

At the same time, we agree that a logarithmic scaling could help to more precisely visualise the size of the microsaccades driving the bias, over and above showing a clear divide between small (micro) vs. large (macro) saccades.

To achieve the best of both worlds, we have now included both linear and log-scaled plots. While linear scaled plots enable to emphasis the key point, and are consistent with the plots that we have used previously in complementary publications, the log-scaled plots help to more precisely visualise the data in the range at which the action happens. We present our updated figure below:

Figure 5. Attentional biasing of saccades is driven by saccades in the microsaccade range. a) Saccade rates as a function of saccade size (y axes) and time after attention-directing cues (x axes) for toward saccades (left), away saccades (middle), and their difference (toward minus away; right). **b)** Overall saccade rates as a

function of saccade size and time after cue onset for trials with attention-directing cues (left), neutral cues (middle), and their difference (attention-direction minus neutral; right). Data were binned and visualised as a function of linearly spaced saccade sizes (top) and logarithmically spaced saccade sizes (bottom). During encoding, items were centred at ± 5 degrees to the left and right of fixation.

- II. 483 the ideas for future studies seems a bit random, so let me add one: it would be interesting to see how the size effect scales with presence vs. absence of a fixation symbol. Would the results be more Spivey-like (I. 238) if a cue like background color or outer frame color were used?

Thank you. This is a great idea, that also resonates with a complementary comment from R3. We now explicitly bring forward this suggestion for another interesting future study:

Page 14, Discussion:

[...] At the same time, we cannot rule out that there may be other settings in which attention shifts could directly trigger microsaccades, such as when there are no concurrent demands on fixation (i.e., in the absence of a fixation marker). Assessing this possibility remains an interesting target for future research.

Reply to final remaining comment of reviewer #2

The authors' response on statistical analysis makes me very sad. One cannot but wonder how many more dozens(!) of years must pass until linear mixed models stop being thought of "sophisticated" and "complex". This is a very minor but important advancement over unregularized repeated measurements test such paired t-test or rm ANOVA (LMM tend to overfit less). The difference is literally in regularized (finite variance, LMM) versus unregularized (infinite variance flat priors, LM) distributions for random effects. Same goes for generalized linear models, is a century enough for logistic regression to finally become widely used on binomial data without converting to proportions and forcing it into an ill-suited linear model analysis such a t-test or ANOVA. throwing out all information about uncertainty in the process? It is also odd to use Cohen's D that works ONLY for normal distributions without checking for that. Cohen's D is effectively a measure of overlap between two normal(!) distributions but if distributions are not normal the formula no longer matches the overlap and is misleading. I understand the allure of linear models, but it only works if your data allows for it, not when you wish it would.

Science relies on the use of appropriate statistics that match outcome variables. Not traditional, not mainstream, not convenient for comparison across different measures of different nature, but one that matches outcome variable distribution and assumptions about generative process. Repeating ill-suited analysis only propagates it further into mainstream (this is literally "if everyone jumped off the bridge" situation).

Thank you for having taken the time to carefully evaluate our revision, and for prompting us to employ more appropriate statistics and statistical reporting. We have now followed your valuable and justified suggestions – and we are grateful to have been given a chance to do so. Specifically:

1. We now explicitly check for normality of the data before reporting any Cohen's D value as a measure of effect size. Where normality was violated (in one case), we now report the outcomes (including the effect size) of the non-parametric Wilcoxon rank sum test instead. We now also state this explicitly in our Methods section.
2. We have now adopted the proposed Linear Mixed Model (LMM) analyses when comparing reaction-time and accuracy scores between conditions, in the way as proposed in the first round of review (using log-transformed RT and using logistic regression for the binary accuracy scores). We are pleased to report that these more appropriate analyses yielded the same statistical inferences of clear differences between conditions.